

# Nitrogen dioxide and formaldehyde measurements from the GEOstationary Coastal and Air Pollution Events (GEO-CAPE) Airborne Simulator over Houston, Texas

Caroline R. Nowlan[1], Xiong Liu[1], Scott J. Janz[2], Matthew G. Kowalewski[2,3], Kelly Chance[1], Melanie B. Follette-Cook[2,4], Alan Fried[5], Gonzalo González Abad[1], Jay R. Herman[6], Laura M. Judd[7], Hyeong-Ahn Kwon[8], Christopher P. Loughner[9], Kenneth E. Pickering[2,10], Dirk Richter[5], Elena Spinei[11], James Walega[5], Petter Weibring[5], and Andrew J. Weinheimer[12]

[1]Harvard-Smithsonian Center for Astrophysics, Cambridge, MA 02138, USA
[2]NASA Goddard Space Flight Center, Greenbelt, MD 20771, USA
[3]Goddard Earth Sciences Technology and Research, Universities Space Research Association, Greenbelt, Maryland, USA
[4]Morgan State University/GESTAR, Baltimore, MD 21251, USA
[5]Institute for Arctic and Alpine Research, University of Colorado, Boulder, CO 80303, USA
[6]University of Maryland, Baltimore County, Baltimore, MD 21201, USA
[7]NASA Langley Research Center, Hampton, VA 23666, USA
[8]Seoul National University, Seoul, Republic of Korea
[9]NOAA Air Resources Laboratory, College Park, MD 20740, USA
[10]University of Maryland, College Park, College Park, MD 20742, USA
[11]Virginia Tech, Blacksburg, VA 24061, USA
[12]National Center for Atmospheric Research, Boulder, CO 80307, USA

**Correspondence:** Caroline Nowlan (cnowlan@cfa.harvard.edu)

**Abstract.** The GEOstationary Coastal and Air Pollution Events (GEO-CAPE) Airborne Simulator (GCAS) was developed in support of NASA's decadal survey GEO-CAPE geostationary satellite mission. GCAS is an airborne pushbroom remote sensing instrument, consisting of two channels which make hyperspectral measurements in the ultraviolet/visible (optimized for air quality observations) and the visible/near-infrared (optimized for ocean color observations). The GCAS instrument

5  participated in its first intensive field campaign during the Deriving Information on Surface Conditions from Column and Vertically Resolved Observations Relevant to Air Quality (DISCOVER-AQ) in Texas in September 2013. During this campaign, the instrument flew on a King Air B-200 aircraft during 21 flights on 11 days to make air quality observations over Houston, Texas. We present GCAS trace gas retrievals of nitrogen dioxide ($NO_2$) and formaldehyde ($CH_2O$), and compare these results with trace gas columns derived from coincident in situ profile measurements of $NO_2$ and $CH_2O$ made by instruments on a

10  P-3B aircraft, and with $NO_2$ observations from ground-based Pandora spectrometers operating in direct sun and scattered light modes. GCAS tropospheric column measurements correlate well spatially and temporally with columns estimated from the P-3B measurements for both $NO_2$ ($r^2 = 0.89$) and $CH_2O$ ($r^2 = 0.54$) and with Pandora direct sun ($r^2 = 0.85$) and scattered light ($r^2 = 0.94$) observed $NO_2$ columns. Coincident GCAS columns agree in magnitude with $NO_2$ and $CH_2O$ P-3B-observed columns to within 10 %, but are larger than scattered light Pandora tropospheric $NO_2$ columns by 33 % and direct sun Pandora

15  $NO_2$ columns by 50 %.




# 1 Introduction

The GEOstationary Coastal and Air Pollution Events (GEO-CAPE) Airborne Simulator (GCAS) is an airborne hyperspectral remote sensing instrument developed in support of future Earth-observing geostationary satellite missions. GCAS was originally developed by NASA Goddard Space Flight Center's (GSFC) Radiometric Calibration and Flight Development Laboratory as a simulator for GEO-CAPE, a NASA decadal survey mission for observing pollution and ocean color from geostationary orbit (Fishman et al., 2012). GCAS is now also a test-bed instrument for the Tropospheric Emissions: Monitoring of POllution (TEMPO) instrument (Chance et al., 2013; Zoogman et al., 2017), which will monitor air quality over North America from a geostationary orbit. TEMPO is the ultraviolet/visible/near-infrared air quality component of GEO-CAPE and is scheduled for launch in the 2019–2021 timeframe. As a satellite airborne simulator, GCAS provides an algorithm development test-bed for GEO-CAPE and TEMPO, serves as a satellite analogue during field campaigns, and will eventually act as a validation instrument when geostationary satellite instruments are on orbit.

GCAS is a pushbroom remote sensing instrument consisting of two spectrometers. The first spectrometer operates in the UV/visible region of the spectrum, where observations can be made of several atmospheric constituents of interest to air quality. The second spectrometer operates in the visible/near-infrared (NIR) for measurements focused on ocean color. In this paper, we focus on air quality observations of nitrogen dioxide ($NO_2$) and formaldehyde ($CH_2O$) using data from the UV/visible channel collected during the Deriving Information on Surface Conditions from Column and Vertically Resolved Observations Relevant to Air Quality (DISCOVER-AQ) campaign in Texas during September 2013. $NO_2$ and $CH_2O$ have spectral absorption signatures in the UV/visible channel and are two core operational data products of future geostationary air quality instruments.

Nitrogen oxides ($NO_x = NO + NO_2$) are of central importance to air quality and atmospheric chemistry. $NO_x$ is involved in the formation of photochemical ozone and fine aerosol particles, with implications for both surface air quality and climate. Both short- and long-term enhanced $NO_2$ concentrations are associated with increased mortality (Hoek et al., 2013; Mills et al., 2015). $NO_x$ emissions can also lead to excess nitrogen deposition (Fowler et al., 2013; Nowlan et al., 2014). Globally, the major sources of $NO_x$ are combustion, lightning and soils. In populated regions, sources are typically dominated by combustion of fuel for transportation and industry. The relatively strong $NO_2$ spectral absorption features at ultraviolet (Yang et al., 2014) and visible (Martin et al., 2002; Boersma et al., 2008; Richter et al., 2011; Bucsela et al., 2013) wavelengths have been used for over two decades to derive global maps of $NO_2$ from several sun-synchronous satellite sensors in low Earth orbit.

Formaldehyde ($CH_2O$) is found in the Earth's atmosphere due to the oxidation of both methane and the non-methane volatile organic compounds (NMVOCs) that result from biogenic and anthropogenic activity and fires (Fried et al., 2008, 2011, 2016a, and references therein). Industrial activity and fires can also be direct sources of $CH_2O$ (Fried et al., 2016b). The absorption signature of $CH_2O$ in the ultraviolet has permitted its detection from the same nadir-viewing satellite instruments that measure $NO_2$ (Chance et al., 2000; De Smedt et al., 2008, 2012; González Abad et al., 2015, 2016). Its short lifetime of ~1.5 – 3 h (around local noon) means that satellite-observed $CH_2O$ can be used as a proxy of NMVOC emissions (Barkley et al., 2008; Zhu et al., 2014; Stavrakou et al., 2015).





Both the air quality and ocean color spectrometers in the GCAS instrument use CCD array detectors to measure solar radiation backscattered from the surface and atmosphere. The pushbroom technique used by GCAS provides data for constructing two-dimensional maps beneath the aircraft, and is also employed by satellite instruments such as the Ozone Monitoring Instrument (OMI) (Levelt et al., 2006) and Ozone Mapping Profiler Suite (OMPS) nadir mapper (Flynn et al., 2014). In these instruments, one axis of the CCD array detector provides spectral information while the other CCD axis provides spatial cross-track information below the aircraft or satellite. The second spatial dimension is provided by the movement of the aircraft or satellite in its flight track.

$NO_2$ amounts over industrial regions and urban areas have been mapped at high spatial resolution by several recently-developed airborne pushbroom sensors (Heue et al., 2008; Popp et al., 2012; Schönhardt et al., 2015; Lawrence et al., 2015; Nowlan et al., 2016; Meier et al., 2017; Tack et al., 2017; Vlemmix et al., 2017; Broccardo et al., 2018). Airborne remote sensing $CH_2O$ measurements have previously been made from aircraft using limb viewing geometry by Airborne Multi-Axis Differential Optical Absorption Spectroscopy (AMAX-DOAS) (Baidar et al., 2013) and by the whiskbroom scanning technique (where the cross-track spatial dimension is provided by mechanical scanning) using the Airborne Compact Atmospheric Mapper (ACAM) (Liu et al., 2015b). Operated by the NASA GSFC Radiometric Calibration and Flight Development Laboratory, ACAM is a precursor instrument to GCAS, and has also been used to measure $NO_2$ and ozone (Liu et al., 2015b; Lamsal et al., 2017). To the best of our knowledge, the GCAS measurements presented here are the first $CH_2O$ published observations from an airborne pushbroom nadir mapper.

GCAS flew in its first field campaign during the DISCOVER-AQ campaign in Texas in 2013. In the following sections, we present and validate trace gas retrievals of $NO_2$ and $CH_2O$ from the GCAS instrument during DISCOVER-AQ Texas. Section 2 describes the GCAS instrument and measurement approach. Section 3 describes the DISCOVER-AQ Texas campaign deployment, measurements from GCAS and relevant ground-based spectrometers and in situ aircraft instruments, and the atmospheric models used in data analysis. Section 4 presents the trace gas retrievals, including the spectral fitting used to derive $NO_2$ and $CH_2O$ slant columns, and air mass factor calculations. Section 5 describes the vertical column results from the campaign. Section 6 presents comparisons of GCAS observations with other coincident observations of $NO_2$ and $CH_2O$.

## 2 The GCAS instrument

The GCAS instrument is a nadir-looking hyperspectral instrument consisting of two Offner spectrometers operating at wavelengths 300–490 nm (UV/visible, air quality channel) and 480–900 nm (visible/NIR, ocean color channel). The instrument has dimensions of 48 cm × 48 cm × 46 cm and a mass of 36 kg. We briefly describe the GCAS instrument below; a more detailed description of the instrument and laboratory characterization can be found in Kowalewski and Janz (2014).

The UV/visible air quality channel consists of a thermo-electrically cooled 1072×1024 CCD detector array, measuring an image with 1072 wavelengths in the spectral dimension and 1024 positions in the spatial dimension across the flight track, with a spectral sampling of 0.2 nm and spectral resolution of 0.57 nm. Polarization sensitivity is reduced by the use of a dual wedge crystal quartz and fused silica depolarizer fitted between the slit and instrument fore-optics. The visible/NIR ocean color



channel uses a 1004×1002 CCD array to collect spectra with a spectral sampling of 0.8 nm and resolution of 2.8 nm, and has an order sorting filter to reduce grating second order effects. The spectrometer units are operated at a temperature of 20°C and are stable to 0.25°C 40 minutes after a nominal take-off to a typical cruise altitude (Kowalewski and Janz, 2014). In addition to the two spectrometers, a video camera is also included in the housing for collecting relevant scene information.

The GCAS instrument fore-optics collect backscattered light below the aircraft through a common fused silica window. The full field of view (FOV) of the air quality channel covers 45° in the cross-track dimension and the instantaneous FOV (IFOV) along the flight track is 0.8 mrad. The ocean color channel full FOV is 70°, with an IFOV of 1.2 mrad. All observations in this study use the UV/visible air quality channel.

Spectra are spatially averaged in post-processing to increase the signal-to-noise ratio for air quality trace gas observations.
NASA GSFC typically produces averaged Level 1B calibrated spectra at 21 cross-track positions, at a spatial resolution of 250 m cross track and 500 m along track from a ~9 km flight altitude, with a resulting signal-to-noise ratio of ~360 at 340 nm and ~540 at 440 nm. GCAS does not have a zenith sky reference measurement capability, unlike the Geostationary Trace gas and Aerosol Sensor Optimization (GeoTASO) (Nowlan et al., 2016) or ACAM (Liu et al., 2015b) airborne instruments also operated by the NASA GSFC.

## 3   DISCOVER-AQ Texas 2013

DISCOVER-AQ (http://discover-aq.larc.nasa.gov/) was a suborbital-class NASA Earth Venture mission consisting of four major field campaigns (Maryland 2011, California 2013, Texas 2013 and Colorado 2014) whose goal was to improve air quality monitoring by satellites. During the campaigns, NASA's King Air B-200 (remote sensing) and P-3B (in situ) aircraft made measurements of trace gases, aerosols and meteorological variables, while balloon-borne, ship-based, mobile, and stationary
instruments collected large amounts of in situ and remote sensing data.

As part of the remote sensing component of DISCOVER-AQ, NASA GSFC deployed the airborne ACAM scanning instrument during the Maryland 2011 (Liu et al., 2015a, b; Lamsal et al., 2017) and California 2013 campaigns, and the GCAS instrument during the Texas 2013 and Colorado 2014 campaigns. Additionally, the first test flights of the GeoTASO airborne instrument, another geostationary airborne simulator, were performed during the Texas (Nowlan et al., 2016) and Colorado
(Crawford et al., 2016) campaigns from the NASA HU-25C Falcon aircraft.

The DISCOVER-AQ Texas campaign took place in September 2013. The campaign aircraft, sondes, and ground-based instruments were based in and around Houston, Texas, an urban area with large emission contributions from both transportation and the petrochemical industry, and air quality often influenced by land-sea breezes. Figure 1 shows the location of the 10 DISCOVER-AQ ground sites with Pandora spectrometers which GCAS overflew, and a day of flight tracks from the King Air
B-200 and P-3B aircraft.





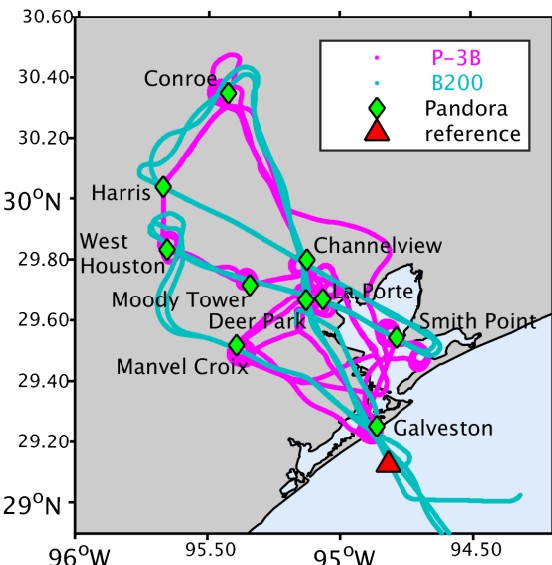

**Figure 1.** Map of Houston area showing sample flight tracks for the King Air B-200 (GCAS) and the P-3B aircraft on 6 September 2013, and ground sites where Pandora spectrometers were located. The red triangle shows the location of the observations used to calculate the GCAS reference spectra.

## 3.1 GCAS observations

During the DISCOVER-AQ Texas campaign in September 2013, the NASA King Air B-200 carried the GCAS instrument for remote sensing of trace gases and aerosols, as well as the NASA High Spectral Resolution Lidar (HSRL) instrument (Hair et al., 2008). The B-200 typically flew at a cruise altitude of ~9 km, with occasional descents to avoid cirrus clouds. Table 1 summarizes the 26 GCAS flights (21 for air quality and 5 for ocean color), which took place on 13 days. Most flights were designed to coincide with P-3B flight paths. The B-200 aircraft was based at Ellington Field in south-east Houston, and typically flew a morning flight, refueled, and then flew an afternoon flight. Each B-200 flight over Houston consisted of two overpasses of nearly the same flight path, so that there are typically four GCAS overpasses of Houston each day. The ocean color flights involved collecting data over the Gulf of Mexico in support of the ocean color component of GEO-CAPE. In this study, we focus only on the air quality flights over the Houston area.

## 3.2 Pandora observations

Total column observations of $NO_2$ were made from 15 ground-based Pandora spectometers viewing in direct sun mode (Herman et al., 2009) at 11 sites during the DISCOVER-AQ Texas campaign. GCAS overflew 14 of these spectrometers at 10 sites, which are summarized in Table 2. Pandora $NO_2$ is determined at a temporal resolution of 90 s using the ratio of direct-sun spectra to a reference spectrum derived by a top-of-the-atmosphere Langley extrapolation using spectra collected on a clear



day with low $NO_2$ (Herman et al., 2009). Spectra are fit from 400–440 nm with $NO_2$ cross sections interpolated to 264 K (Vandaele et al., 1998) and $O_3$ at 225 K (Brion et al., 1993). At solar zenith angles (SZA) less than 80°, the observed direct sun slant column is converted to vertical total column using a simple geometric air mass factor (Herman et al., 2009). Pandora $NO_2$ measurements have a nominal precision of $2.7 \times 10^{14}$ molecules cm$^{-2}$ and accuracy of $2.7 \times 10^{15}$ molecules cm$^{-2}$.

Pandora observations with fitting RMS<0.005 and relative error <10 % are included in this study, to exclude possible cloud-contaminated measurements.

Pandoras also operated in multi-axis sky scanning mode (MAX-DOAS) measuring lower tropospheric $NO_2$ distribution and tropospheric columns at the La Porte, Moody Tower and Smith Point sites. Pandora head sensors sequentially pointed at 1, 2, 3, 4, 6, 8, 10, 15, 20, 30, 40, and 90° elevation angles from the horizon with a field of view of 1.6°. Azimuth angles were

chosen to ensure an unobstructed view down to the horizon and were 320° from north at La Porte, 45° at Moody Tower and 270° at Smith Point. Differential slant column densities of $NO_2$ and $O_2-O_2$ within a single scan were calculated using a zenith sky reference spectrum. A temperature dependent $NO_2$ absorption cross section (linear and constant terms) and Ring, $H_2O$ and $O_2-O_2$ cross sections (see Table 3 for references) were used in MAX-DOAS fitting window 425–490 nm. The profile inversion was performed using the maximum a posteriori optimal estimation method (Rodgers, 2000) with aerosol

and gas weighting functions calculated using VLIDORT 2.7 (Spurr, 2008). Tropospheric columns were also estimated using a geometrical approach using $NO_2$ and $O_2-O_2$ columns derived from 15° elevation angle measurements when inversions failed.

The Pandora dataset contains observations from two Pandora instruments placed at the Moody Tower site at the University of Houston, 70 m above the surface. We correct for the column in the bottom 70 m of the atmosphere using in situ observations at the base and top of the towers collected every 5 min by the University of Houston following Nowlan et al. (2016). The in

situ measurements indicated that $NO_2$ within these altitudes was usually well-mixed at the overpasses. This correction varies in magnitude from $0.3 \times 10^{15}$ molecules cm$^{-2}$ to $3.7 \times 10^{15}$ molecules cm$^{-2}$ for different GCAS overpasses.

### 3.3 P-3B aircraft observations

The P-3B aircraft carried a suite of in situ instruments for profiling the atmosphere during the campaign. Profiles were collected during aircraft spirals near eight DISCOVER-AQ ground sites, with each site typically overflown two or three times each day.

Depending on the site and flight, the aircraft typically flew between a lowermost altitude of 0–300 m and an uppermost altitude of 3.5–5 km. The typical radius of a spiral was 4–5 km.

The National Center for Atmospheric Research's (NCAR) chemiluminescence instrument (P-CL) (Ridley and Grahek, 1990) measured in situ $NO_2$ concentrations from the P-3B. P-CL observations of $NO_2$ have uncertainties of 0.02 ppbv in precision and 10 % in accuracy.

The NCAR Differential Frequency Generation Absorption Spectrometer (DFGAS) (Weibring et al., 2006, 2007) measured in situ $CH_2O$ concentrations from the P-3B. The DFGAS instrument collects data with a temporal resolution of 1 second, with a 15-second background zero air addition period every 60 to 120 seconds. This addition captures and removes both inlet/sample cell $CH_2O$ outgassing as well as optical noise. For a typical spiral, the temporal resolution translates to a vertical resolution of



approximately 5 m. The 1-second measurements have a precision of ~0.08 ppbv (upper limit) and an estimated accuracy of 4 % at the $1\sigma$ level.

## 3.4 Model simulations

This study uses model simulated trace gas profiles for radiative transfer calculations in order to determine vertical column
densities from observed slant column densities. Tropospheric simulations are performed with the Environmental Protection Agency's (EPA) Community Multiscale Air Quality (CMAQ) version 5.0.2 modeling system (Byun and Schere, 2006) over the campaign domain at a spatial resolution of $4 \times 4\,\mathrm{km}^2$ and a temporal resolution of 20 minutes. The model has 45 vertical levels from the surface to 50 hPa. CMAQ simulations are driven by offline meteorology from the Advanced Research Weather and Forecasting (WRF-ARW) model (Skamarock et al., 2008) via the Meteorology-Chemistry Interface Processor (MCIP)
(Otte and Pleim, 2010). Loughner and Follette-Cook (2015) describe the CMAQ and WRF modeling approach used for the DISCOVER-AQ Texas campaign in detail.

Stratospheric $NO_2$ profiles used in the study are estimated using the PRATMO chemical box model (Prather, 1992; McLinden et al., 2000) from simulated profiles provided as a function of month, solar zenith angle and latitude. Stratospheric ozone profiles are from the September 2013 monthly climatology derived at $1° \times 1°$ from the OMI ozone profile product (Liu et al.,
2010) up to 0.3 hPa.

## 4  GCAS trace gas retrievals

The GCAS vertical column density retrieval uses a two-step approach. First, we derive the slant column density (SCD) by directly fitting a modeled spectrum to the observed spectrum, starting from a reference spectrum derived from observations over an unpolluted area. Second, we convert SCD to a vertical column density (VCD) using an air mass factor (AMF) that
represents the path of light through the atmosphere based on the viewing geometry and radiative transfer calculations.

The GCAS trace gas retrieval algorithms used in this paper are derived from the Smithsonian Astrophysical Observatory (SAO) trace gas algorithms originally developed for GOME, and since applied to GOME-2, SCIAMACHY, OMI, OMPS and GeoTASO for a range of trace gases (Chance, 1998; Chance et al., 2000; Sioris et al., 2004; Nowlan et al., 2011; Chan Miller et al., 2014; Wang et al., 2014; González Abad et al., 2015, 2016; Nowlan et al., 2016). These algorithms are also the basis
for the TEMPO trace gas retrieval algorithms. A separate slant column trace gas product at $350\,\mathrm{m} \times 1000\,\mathrm{m}$ was provided by the GCAS instrument team at NASA GSFC to the DISCOVER-AQ data archive shortly after the campaign using the publicly-available QDOAS spectral fitting package (http://uv-vis.aeronomie.be/software/QDOAS/) and preliminary calibrated spectra. This product is not examined in the current study.

Nowlan et al. (2016) compared preliminary SAO GCAS $NO_2$ slant columns with GeoTASO slant columns within 10 min-
utes and 500 m from four coincident flights during the DISCOVER-AQ Texas campaign (13, 14, 18 and 24 September) at a resolution of $250\,\mathrm{m} \times 500\,\mathrm{m}$. Overall, slant columns agreed well ($r = 0.81$, $N = 77320$), with GCAS lower than GeoTASO by ~6 %. The current GCAS retrieval algorithm used in this study is similar to the previous algorithm, but the slant column





retrieval uses a separate reference spectrum for each cross-track position and a cross-track dependent instrument line shape, so that the results no longer require a cross-track bias correction. The new $NO_2$ and $CH_2O$ products also include improved georegistration.

## 4.1 Calibration

We perform spectral fitting to derive slant columns using radiometrically-calibrated spectra (Level 1B), which are geolocated and derived from raw (Level 0) data using characterization data collected in the laboratory before the campaign, as described in detail by Kowalewski and Janz (2014). The first-guess wavelength calibration was determined from spectra collected in the laboratory using mercury-argon, cadmium, neon and krypton discharge lamps as sources. The pre-flight slit function shape and width as functions of cross-track position, wavelength and temperature were also determined using a tunable-laser with

an integrating sphere. These laboratory tests indicated the instrument's spectral shift is ~0.004 nm and the change in the slit function's full width at half maximum (FWHM) is less than 0.0013 nm within the instrument's thermal stability range of ±0.25°C and nominal operating temperature of 20°C. The absolute radiometric calibration was determined using a NIST-calibrated integrating sphere with an uncertainty of 3 % and stable to 1 %. Polarization sensitivity was determined using an integrating sphere and polarizer, and is on the order of 1–3 %, depending on field angle (Kowalewski and Janz, 2014).

We further refine the instrument spectral registration and slit function calibration using spectra collected during the Texas flights, following our calibration approach previously applied to GOME, GOME-2, OCO-2, ACAM and GeoTASO (Liu et al., 2005; Cai et al., 2012; Liu et al., 2015b; Nowlan et al., 2016; Sun et al., 2017a, b). As a first step in the spectral fitting, we simultaneously derive a wavelength dispersion and slit function shape by fitting a reference spectrum to a high spectral resolution solar atlas (Chance and Kurucz, 2010). This is similar to the approach employed in our satellite retrievals, but as the

airborne nadir reference spectrum contains atmospheric features (which are not present in a satellite-observed exo-atmospheric reference), we also simultaneously fit preliminary amounts of the atmospheric molecular absorbers listed in Table 3 and the Ring effect (rotational Raman scattering) to account for these spectral features (Liu et al., 2015b; Nowlan et al., 2016).

We determine a separate wavelength dispersion and slit function shape for each of the 21 cross-track positions. The wavelength dispersion is determined by fitting the coefficients in a 5th-order ($NO_2$) or 7th-order ($CH_2O$) polynomial that represents

the wavelength as a function of detector pixel. For $NO_2$, we model the slit function using an asymmetric super Gaussian (Beirle et al., 2017). For $CH_2O$, we fit parameters describing the shape and width of an asymmetric Gaussian (Cai et al., 2012; Nowlan et al., 2016). While the super Gaussian works well for $NO_2$, it results in a very small increase (~5 %) in fitting residuals for $CH_2O$ over an asymmetric Gaussian, possibly due to the presence of a double shoulder to one side of the slit function shape, as measured in the laboratory at wavelengths less than 380 nm (Kowalewski and Janz, 2014).

## 30 4.2 Slant column retrieval

We determine $NO_2$ and $CH_2O$ slant columns using least-squares minimization to directly fit a modeled radiance spectrum $\mathbf{F}(\mathbf{x}, \mathbf{b})$ to our observed radiance spectrum. The modeled spectrum is a function of pre-determined model parameters $\mathbf{b}$ and





the retrieved state vector $\mathbf{x}$. The modeled spectrum is represented by

$$F(\lambda) = ([x_a I_0(\lambda) + b_u(\lambda)x_u + b_r(\lambda)x_r]e^{-\sum_i b_i(\lambda)x_i}) \sum_j (\lambda - \bar{\lambda})^j x_j^{SC} + \sum_k (\lambda - \bar{\lambda})^k x_k^{BL}. \tag{1}$$

In this equation, $I_0$ is a reference spectrum determined from clean nadir observations, scaled by a retrieved intensity parameter $x_a$ (which represents reflectivity factors such as surface albedo or clouds). The derivation of the reference is discussed in

Section 4.3. The term $b_u(\lambda)$ describes a correction for spectral undersampling (Chance et al., 2005), while $b_r(\lambda)$ represents the effects of rotational Raman scattering (Chance and Spurr, 1997). The retrieved differential slant columns are represented by $x_i$ and include the gases listed in Table 3. Their absorption cross sections, as listed in Table 3, convolved with the instrument line shape and corrected for the "$I_0$ effect" (Aliwell et al., 2002), are included as $b_i(\lambda)$. In addition, the retrieval also determines scaling (of order $j$) and baseline (of order $k$) wavelength-dependent polynomial coefficients ($x^{SC}$ and $x^{BL}$) that represent low

frequency wavelength-dependent effects from surface reflectivity, molecular scattering, aerosols and instrument artifacts.

## 4.3 Reference spectrum

Each trace gas retrieval uses reference spectra determined from nadir observations over a clean area. We determine a mean reference spectrum for each of the 21 cross-track positions by averaging 40 spectra at $250\,\mathrm{m} \times 500\,\mathrm{m}$ resolution for each cross-position from a cloud-free and clean area over the Gulf of Mexico during the 6 September afternoon flight. This location and

date were chosen after CMAQ simulations and preliminary retrievals of $NO_2$ and $CH_2O$ predicted relatively low columns of those trace gases. In addition, we found that the use of a reference collected before the instrument was thermally stable (within the first ~40 minutes of a flight) resulted in cross-track biases in the $CH_2O$ retrieval. As observations over the relatively clean Gulf are often collected in the period soon after take-off, this constraint limited the availability of a suitable reference to a reference spectrum taken late in the flight, close to landing, and with a relatively high solar zenith angle (58°).

We use a single reference spectrum at each cross-track position for the entire campaign, instead of a daily or higher frequency reference, to ensure that all days during the campaign have the same background correction applied for the reference spectrum. Due to the use of a nadir reference, the retrieved differential slant columns must be corrected by the reference background column derived from the model to produce an effective tropospheric column (discussed in further detail in Section 4.5). We find that the use of a single reference for the campaign removes day-to-day relative background biases in the GCAS column

that can result from uncertainties in daily modeled columns, and improves the daily consistency of background $CH_2O$ in the in situ P-3B $CH_2O$ comparison discussed later in Section 6.1. The use of a single versus daily reference spectrum has little effect on the $NO_2$ validation.

## 4.4 $NO_2$ and $CH_2O$ fitting

The $NO_2$ and $CH_2O$ slant column density retrievals use the fitting parameters summarized in Table 3. $NO_2$ is fit at wavelengths

420–465 nm with an $NO_2$ absorption cross section at 294 K. The $NO_2$ retrieval also simultaneously fits $O_3$ at two temperatures, as well as $H_2O$ vapor and $O_2-O_2$, which all have spectral absorption features in the $NO_2$ wavelength fitting window. The $CH_2O$ retrieval is performed at 328.5–356.5 nm, and simultaneously fits $NO_2$, $O_3$, BrO and $O_2-O_2$. Both retrievals also




fit the undersampling correction, Ring spectrum, a 5th-order scaling polynomial, and a 4th-order baseline polynomial. Each retrieval also determines a wavelength shift that represents the relative difference in the detector pixel to wavelength registration between the radiance and reference spectra.

## 4.5 Conversion to vertical column

For air quality applications, we are interested in the vertical column density, $V$, of the absorber ($NO_2$ or $CH_2O$) in the troposphere. The vertical column density can be derived from the slant column density, $S$, using an air mass factor, $A$, which describes the mean light path through the atmosphere by

$$V = \frac{S}{A}. \tag{2}$$

In practice, the retrieval algorithm determines a differential slant column $\Delta S$, which is the difference between the slant column $S$ of the absorber in the spectrum of interest, and the slant column $S_R$ in the reference spectrum. Each of these slant columns is the sum of the slant column of absorber in the light path above ($\uparrow$) and below ($\downarrow$) the aircraft, so that

$$\Delta S = (S^\downarrow + S^\uparrow) - (S_R^\downarrow + S_R^\uparrow). \tag{3}$$

In terms of the air mass factor and vertical column, the vertical column below the aircraft can then be expressed as

$$V^\downarrow = \frac{\Delta S - V^\uparrow A^\uparrow + V_R^\downarrow A_R^\downarrow + V_R^\uparrow A_R^\uparrow}{A^\downarrow}, \tag{4}$$

where the vertical columns $V^\uparrow$, $V_R^\downarrow$ and $V_R^\uparrow$ are typically determined from a model. Because the flight altitude of 9 km is well above the majority of tropospheric $NO_2$ and $CH_2O$, we refer to $V^\downarrow$ and $V^\uparrow$ as the trospospheric and stratospheric trace gas columns.

### 4.5.1 Air mass factor calculation

We calculate the air mass factors on a scene-by-scene basis using the formulation of Palmer et al. (2001) and Martin et al. (2002) with the VLIDORT radiative transfer model (Spurr, 2006, 2008). In this approach, the radiative transfer model provides scattering weights $w$ as a function of altitude $z$. The scattering weights describe the sensivity of the measurement to the different altitude layers and are a function of the viewing geometry, ozone profile, aerosol and molecular scattering, and surface reflectance. These can be used with shape factor $s$, which is the normalized partial column $n$ of the trace gas at at each altitude layer:

$$s(z) = \frac{n(z)}{\int_z n(z)\mathrm{d}z}. \tag{5}$$

The AMF is defined as

$$A = \int_z w(z)s(z)\mathrm{d}z. \tag{6}$$



The air mass factor below the aircraft $A^\downarrow$ is calculated from the surface $z_0$ to the aircraft altitude $z_{ac}$ as

$$A^\downarrow = \int\limits_{z_0}^{z_{ac}} w(z)s(z)\mathrm{d}z, \tag{7}$$

while the air mass factor above the aircraft $A^\uparrow$ is determined from the aircraft altitude to the top of the atmosphere at $z_{TOA}$, with

$$A^\uparrow = \int\limits_{z_{ac}}^{z_{TOA}} w(z)s(z)\mathrm{d}z. \tag{8}$$

### 4.5.2 Radiative transfer calculations

We use the radiative transfer algorithm to determine scattering weights in 56 vertical layers. These include the 45 CMAQ layers up to ~19 km and 11 additional layers to 0.3 hPa. We use the MODIS BRDF (bidirectional reflectance distribution functions) gap-filled MCD43GF V005 Band 3 product (Schaaf et al., 2002; Sun et al., 2017) to represent surface reflectance in the VLIDORT model. The MODIS Band 3 product is derived at 470 nm. While this is close to the $NO_2$ fitting window, there currently exists no BRDF climatology at shorter wavelengths. We determine effective BRDFs at 442 nm ($NO_2$) and 342 nm ($CH_2O$) by scaling the BRDF functions by the ratio of the $0.5° \times 0.5°$ monthly OMI Earth Surface Reflectance Climatology product (OMLER) (Kleipool et al., 2008) at either 442 nm or 342 nm to its value at 470 nm. These results are typically within a few percent of the results derived using a black-sky/white-sky approach to estimate surface reflectance (McLinden et al., 2014).

Figure 2 shows profiles for 1) a sample polluted observation at the Moody Tower site in downtown Houston and 2) the reference spectrum. For the AMF calculation, the shape factors are derived from the model profiles shown in Figures 2a and 2c and then applied to the corresponding scattering weights. Differences in the scattering weights of the reference and Moody Tower observations at higher altitudes are mainly driven by differences in the solar zenith angles. The smaller $CH_2O$ scattering weights near the surface relative to those of $NO_2$ indicate the relatively lower sensitivity of the observations to near-surface $CH_2O$. This is due primarily to the wavelength dependency of the AMF, as stronger Rayleigh scattering and ozone absorption at shorter wavelengths decreases the measurement sensivity to lower altitudes.

### 4.6 Cloud flagging

Only cloud-free measurements are used in this study, and the radiative transfer calculations assume cloud-free conditions. Unlike the case of satellite observations with footprints on the order of tens of square kilometers, GCAS observations are of sufficiently high spatial resolution that cloudy pixels can be discarded without loss of a significant amount of data. We flag as cloudy any pixel that has a mean radiance in the $NO_2$ fitting window over a threshold of $2 \times 10^{13}\,\mathrm{photons\,cm^{-2}\,nm^{-1}\,s^{-1}\,sr^{-1}}$, which is typically only exceeded in the case of a bright cloud. The Ring scattering parameter retrieved simultaneously with $NO_2$ and a color index (the radiance ratio at wavelengths 320 to 440 nm) are also used to flag less bright pixels where clouds likely occur (Wagner et al., 2014).





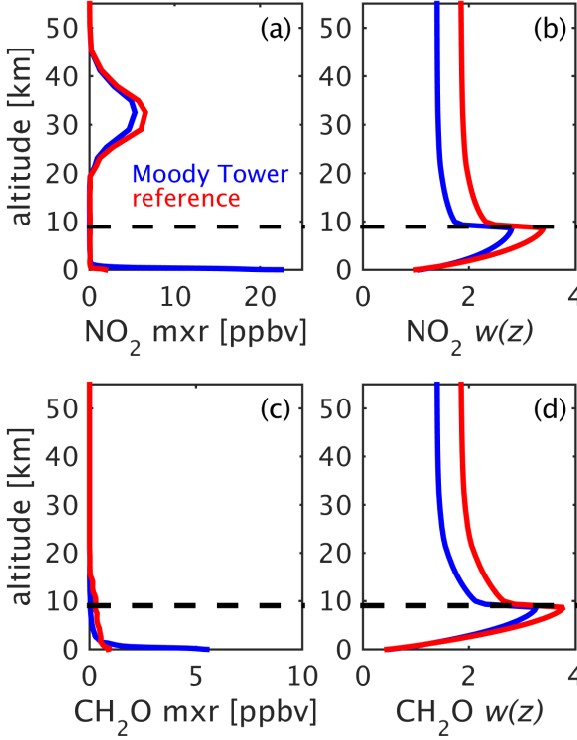

**Figure 2.** Sample mixing ratio (mxr) and scattering weight ($w(z)$) profiles used in the GCAS AMF calculations. The reference spectrum profiles are taken from the 6 September afternoon flight over the Gulf of Mexico at average location 29.126°N, 94.818°W, 17:04 LT (local time = UTC time - 5 hours) with SZA=58.0°, VZA=10.5°. The profiles at the Moody Tower site in downtown Houston are from the 25 September morning flight at 10:56 LT with SZA=45.0°, VZA=10.7°. The estimated surface reflectivities at 442 nm (NO$_2$) and 342 nm (CH$_2$O) are 0.04 and 0.05 at the reference location and are both 0.07 at Moody Tower. The dashed black line indicates the aircraft flight altitude.

## 4.7 Trace gas uncertainties

Uncertainties in the vertical column density result from uncertainties in 1) the slant column fitting and 2) the air mass factor calculation.

### 4.7.1 Slant column uncertainties

5  The slant column fitting uncertainty on a single observation is dominated by the random noise in the spectrum. Over a typical day, the mean fitting uncertainty in a NO$_2$ differential slant column at $250\,\text{m} \times 500\,\text{m}$ resolution is $1.3\times10^{15}$ molecules cm$^{-2}$, including all solar zenith angles in the morning and afternoon flights. The typical fitting uncertainty in a CH$_2$O differential slant column is $2.5 \times 10^{16}$ molecules cm$^{-2}$. After AMFs are applied, typical mean vertical column precisions are $1 \times 10^{15}$ molecules cm$^{-2}$ for NO$_2$ and $1.9 \times 10^{16}$ molecules cm$^{-2}$ for CH$_2$O. The precision requirements of the TEMPO instrument





are $1 \times 10^{15}$ molecules cm$^{-2}$ for NO$_2$ and $1 \times 10^{16}$ molecules cm$^{-2}$ for CH$_2$O (Zoogman et al., 2017). (Note that the signal-to-noise is higher at CH$_2$O wavelengths relative to that at NO$_2$ wavelengths for TEMPO, which is the opposite of GCAS.) CH$_2$O in particular is noisy at the provided GCAS spatial resolution of $250\,\text{m} \times 500\,\text{m}$, with enhanced CH$_2$O columns often on the order of the retrieval precision. GCAS CH$_2$O must be spatially averaged to meet the TEMPO precision requirement and

to improve the detection limit in order to observe polluted columns over Houston. As a result, later in this paper we present CH$_2$O maps at $1\,\text{km}^2$ resolution. It should be noted that even at precisions of $1 \times 10^{16}$ molecules cm$^{-2}$, CH$_2$O columns from satellite instruments like OMI typically must be temporally averaged to resolve local CH$_2$O features (e.g., Marais et al., 2014; Zhu et al., 2014).

Additional errors in NO$_2$ slant column retrievals can also result from the use of an NO$_2$ cross section at a single temperature

(Boersma et al., 2004). The profile-weighted effective temperature of NO$_2$ during the Houston campaign in polluted observations was typically within a few degrees of the 294 K cross section temperature, resulting in an expected bias within 1–2 % in the tropospheric slant column. The stratospheric slant column may be biased by ~15 % due to its colder temperature, but the influence of this uncertainty is minimized by the use of a nadir reference spectrum, resulting in a possible systematic bias on the order of $4 \times 10^{14}$ molecules cm$^{-2}$ (an uncertainty of 1–2 % for polluted pixels). Uncertainties in the laboratory cross

sections introduce additional uncertainties in the slant columns of 2 % for NO$_2$ (Boersma et al., 2004) and 5 % for CH$_2$O (Chance and Orphal, 2011).

### 4.7.2 Air mass factor uncertainties

The air mass factor uncertainties in cloud-free satellite observations are typically dominated by uncertainties in the surface albedo, trace gas profile shape and aerosols (Boersma et al., 2004). A recent study by Lorente et al. (2017) found an average

AMF structural uncertainty of 42 % in polluted observations and 31 % in unpolluted regions when different retrieval groups used different inputs to NO$_2$ AMF calculations; the most significant impacts overall were from differences in surface albedo, cloud parameters and trace gas profile inputs.

MODIS BRDF comparisons with aircraft observations of the surface indicate an uncertainty in the MODIS BRDF product of 20 % for both accuracy and precision (Román et al., 2011) at GCAS spatial resolutions. We estimate the impact of those

uncertainties from the MODIS surface BRDF on our individual AMFs to be 10 % for polluted observations and 5 % for clean observations. Wang et al. (2010b) showed that the use of the Lambertian approximation in the derivation of the MODIS products may result in surface reflectance underestimation of 0.008 by MODIS in the green bands. This surface bias on average could cause the GCAS AMF to be underestimated (and the resulting trace gas column to be overestimated) by ~10 %.

The radiative effects of aerosols are not typically included in operational satellite retrieval trace gas AMFs, except as an

implicit component of the cloud fraction, and we have not included aerosols in the current study. In reality, the presence of aerosols can increase or decrease the AMF, with effects depending on aerosol type and altitude (Leitão et al., 2010; Lin et al., 2014; Chimot et al., 2016; Kwon et al., 2017). When scattering aerosols are in the boundary layer, for example, the backscattered light path increases the radiative sensitivity (an enhancement effect), resulting in an increase in the AMF. Ignoring these aerosols in the radiative transfer calculation will cause the retrieved column to be overestimated. When scattering aerosols





are aloft, the radiative sensitivity decreases near the surface (a shielding effect), resulting in a decrease in the AMF. Absorbing aerosols aloft or at the altitude of the trace gas can decrease the measurement sensivity by reducing the number of photons backscattered to the instrument, thereby reducing the AMF. Even when aerosols are considered, assumptions about aerosol optical properties and profiles can cause large uncertainties; Lorente et al. (2017) found different aerosol corrections used

by different research groups introduced an average uncertainty of 50 % for polluted satellite observations with high aerosol loading.

Aerosol optical depth (AOD) measured by the HSRL lidar on the B-200 (Sawamura et al., 2017) showed aerosols varying day-to-day, along the flight track and with altitude during the DISCOVER-AQ Texas campaign. The beginning of the campaign saw moderate AOD on the order of 0.2–0.3 (532 nm), often with a smoke plume at altitudes 2–4 km which sometimes merged

with aerosols from lower layers later in the day. Observed AODs rose sharply on 14 September, with AODs in excess of 0.7 in some areas. Aerosol loading from 18 September onwards was relatively low (<0.15) and primarily located near the surface with occasional AOD reaching 0.25 at some points along the flight track. A full assessment of the effects of aerosols on the AMF is beyond the scope of this paper and the subject of ongoing work, but our simulations with typical AOD profiles from the HSRL lidar show a potential overestimation of the column of 10–30 % for individual polluted pixels when scattering aerosols

in the PBL are ignored, and a potential 15 % underestimation of the column when the smoke layer aloft is ignored. These results are consistent with Lin et al. (2014), whose satellite biases are typically within $\pm 25$ % due to the neglect of aerosols at these AODs.

Nowlan et al. (2016) previously compared mean profile shapes from the P-CL observations and the CMAQ simulations for the eight core ground sites during DISCOVER-AQ Texas; mean differences were typically within 20 % for individual sites.

Individual total column observations can vary by >100 % (Nowlan et al., 2016), with differences mostly resulting from the small scale features of $NO_2$ plumes, which are difficult to resolve with model resolution. We also estimate an uncertainty of 30 % in the $NO_2$ stratospheric column, based on PRATMO comparisons with the Optical Spectrograph and InfraRed Imaging System (OSIRIS) limb sounder (Bourassa et al., 2011). Previous comparisons of DISCOVER-AQ Texas CMAQ $CH_2O$ 1-km simulations with P-3B DFGAS observations showed agreement between the model and observations for most days of

the campaign (Fried et al., 2016b). The average of daily mean biases indicated a low bias of CMAQ relative to DFGAS of $-0.44 \pm 0.39$ ppbv in the PBL and $-0.32 \pm 0.40$ ppbv overall ($-11.8 \pm 15.7$%) over all days, excluding 25 September. September 25 was a unique day characterized by very large $CH_2O$ levels of up to 25 ppbv as measured by the DFGAS instrument on the P-3B in the boundary layer over pretrochemical facilities in Houston, and up to 33 ppbv downwind over Galveston Bay and Smith Point later in the day due to photochemical processing (Fried et al., 2016b).

Souri et al. (2018) calculated GCAS $NO_2$ vertical columns independently for our derived slant columns and found a mean tropospheric AMF over all days of $1.26 \pm 0.32$. This compares closely with our mean AMF of $1.29 \pm 0.27$. Their inputs included MODIS BRDF for surface reflectance, GEOS-Chem modeled stratospheric profiles, and an independently-run CMAQ simulation whose aerosol fields were used to determine aerosol optical depths for input to the VLIDORT model. The similar AMF from a separate study suggests a low structural uncertainty in AMF calculations using currently available ancillary information.



## 5   Vertical column results

Figures 3 and 4 show examples of retrieved $NO_2$ and $CH_2O$ tropospheric vertical columns for two consecutive days during the campaign, and illustrate both the day-to-day and hourly variabilities observed in $NO_2$ and $CH_2O$ columns. The 24 September day is more typical of columns measured during the campaign in terms of magnitude. The 25 September flights show the

largest pollution episode of the campaign; this case study has been previously examined in model and in situ measurement studies of ozone, $NO_x$ and $CH_2O$ (Loughner and Follette-Cook, 2015; Souri et al., 2016; Fried et al., 2016b; Mazzuca et al., 2017; Pan et al., 2017).

In general, the largest $NO_2$ columns are seen in morning flights during all days of the campaign. On a given day, the location of the peak columns varies with overpass time and meteorology, and the largest $NO_2$ columns are typically concentrated

over central Houston (close to Moody Tower), in the vicinity of the Houston Ship Channel industrial area (Pandora sites Channelview, Deer Park and La Porte) or sometimes along the more suburban flight track to the west of and over Manvel Croix, which is the case for morning overpasses on 6 and 13 September (Nowlan et al., 2016). Individual $NO_2$ plumes can also often be observed from individual industrial facilities.

Formaldehyde observations are noisier, but enhanced $CH_2O$ columns are clearly observable on some days when data are

spatially averaged. In particular, 4 and 25 September show the largest $CH_2O$ enhancements, with peak values on the order of $5 \times 10^{16}$ molecules $cm^{-2}$ at 1 $km^2$ resolution. Figure 4 shows the significant enhancement in $CH_2O$ near the Houston Ship Channel industrial area on 25 September. High levels of $CH_2O$ measured in situ on this day have been previously attributed to increased emissions from the ExxonMobil Baytown Complex (located 9 $km$ to the northeast of La Porte) and the industrial Channelview/Deer Park region (Fried et al., 2016b). Several other days exhibit enhanced backgrounds over land, with the

largest values of $CH_2O$ columns on these days to the north of Houston over the Conroe region, potentially from biogenic sources as well as transport of $CH_2O$ and its precursors. These days with large background $CH_2O$ highlight the importance of using the clean reference over the water, where background $CH_2O$ is typically lower than over land.

## 6   Comparisons with coincident measurements

In this section, we compare GCAS observations from all days with coincident observations from Pandoras and the P-3B

aircraft. Figures 5 and 6 show enlarged views of $NO_2$ and $CH_2O$ observations over the downtown and Ship Channel regions of Houston on 25 September, along with coincident Pandora ground site observations and the P-3B flight track nearest in time. These figures illustrate the typical coverage of P-3B spirals relative to GCAS swaths, as well as the near-surface air mass measured by Pandora ground-based instruments.

### 6.1   P-3B airborne in situ measurements

Figure 7 shows regression plots for the GCAS $NO_2$ and $CH_2O$ columns as a function of those derived from the respective in situ instruments on the P-3B aircraft, with statistics computed using a bivariate regression where the residuals in both



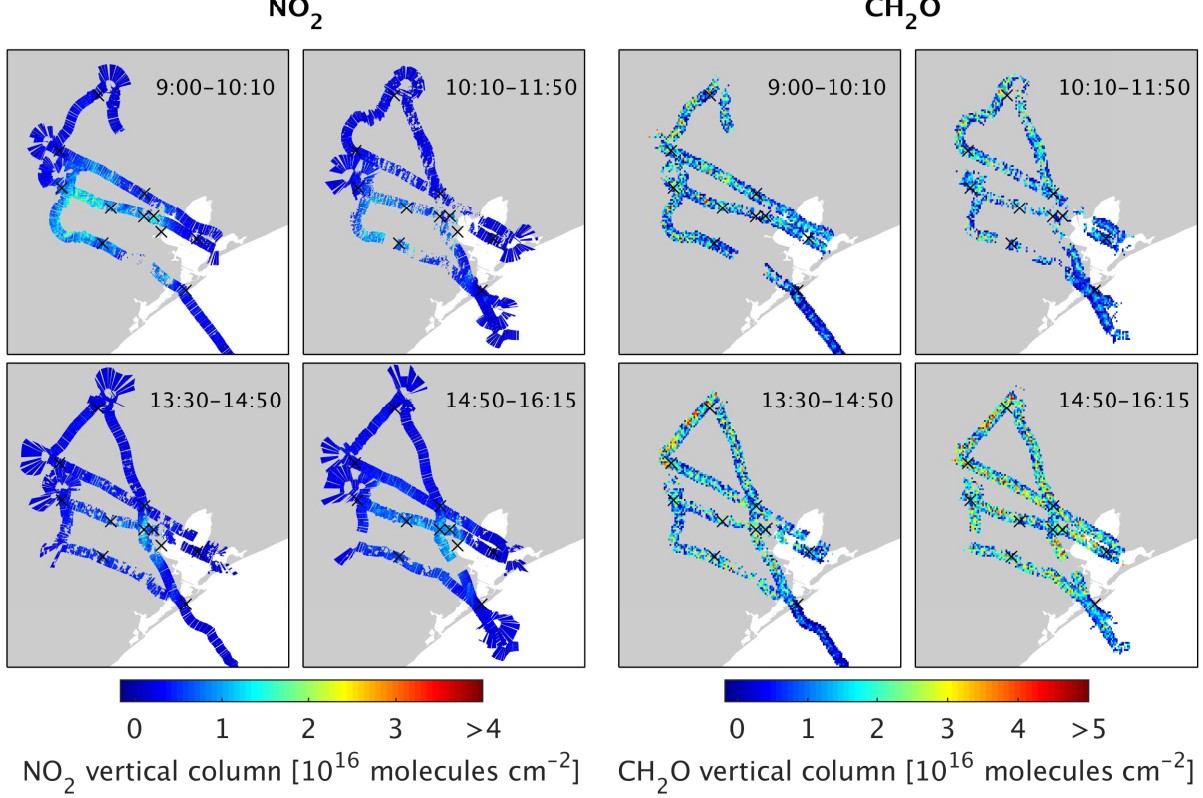

**Figure 3.** Tropospheric $NO_2$ and $CH_2O$ vertical columns measured by GCAS over Houston on 24 September 2013. $NO_2$ observations are at ~250 m × 500 m resolution and $CH_2O$ columns are at $0.01° × 0.01°$ (~1 km$^2$) resolution. Times are local time. Black crosses indicate ground sites.

measurements are minimized. Each P-3B column is calculated by integrating the $NO_2$ or $CH_2O$ partial columns derived from observed mixing ratios over the altitude of the spiral. The lowest altitude of each P-3B spiral varies by location. At Deer Park, Galveston and West Houston, the mean minimum spiral altitude is ~20–40 m, while Conroe and Smith Point spirals typically go as low as ~130 m. At Channelview, Manvel Croix, and Moody Tower, the lowest spiral altitude is typically ~300 m. To

5  determine the $NO_2$ profile below the lowest P-3B altitude, we estimate the P-3B mixing ratio below the aircraft following Lamsal et al. (2014), by extrapolating the mixing ratio at the lowest aircraft altitude to the surface using the vertical gradient from the CMAQ model at altitudes below the spiral. $CH_2O$ DFGAS mixing ratios below the spiral are extrapolated to the ground from the lowest mixing ratio in the bottom 100 m of the spiral, as described by Fried et al. (in preparation).

The GCAS column is calculated by averaging all GCAS columns within 1 h and 5 km of a spiral center. We exclude

10  spirals where there are less than 30 GCAS observations within the coincident area. The modeled $NO_2$ column above the top P-3B spiral altitude is subtracted from the retrieved GCAS tropospheric $NO_2$ column (~$3 \times 10^{14}$ molecules cm$^{-2}$ on average).





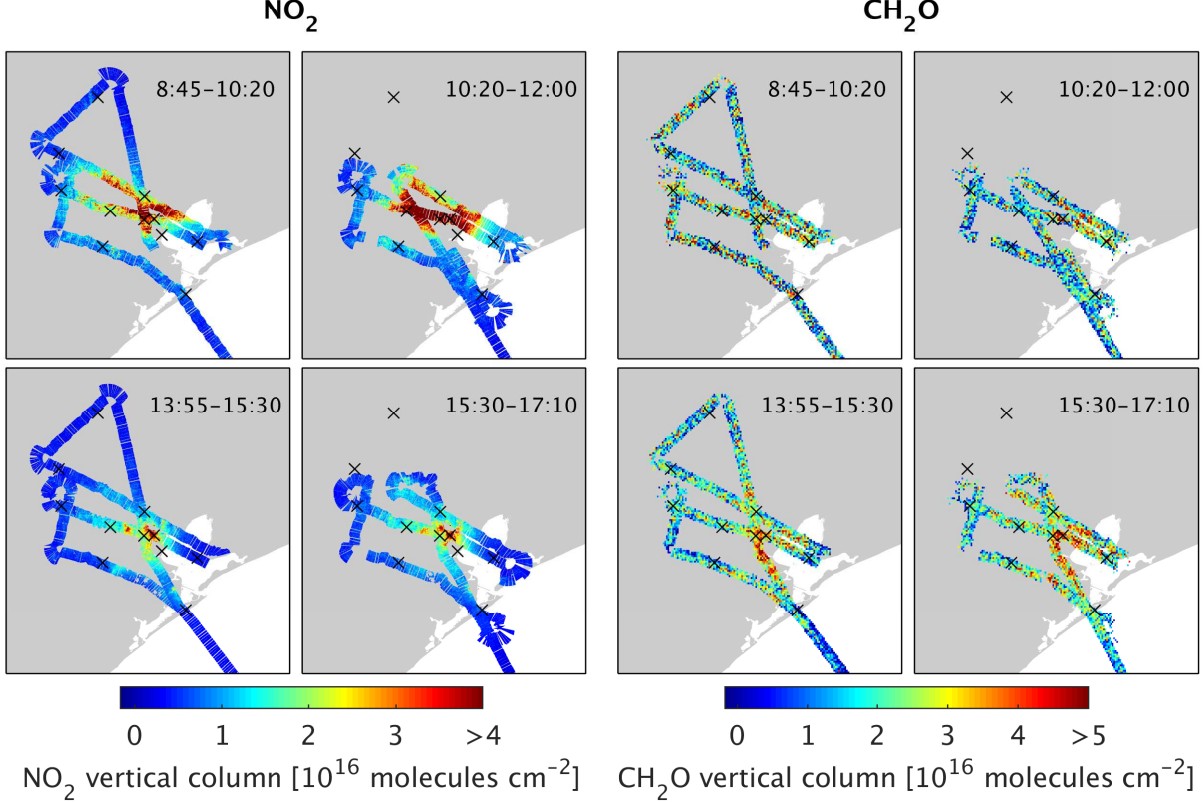

**Figure 4.** Same as Fig. 3 but for 25 September 2013.

According to the CMAQ simulations, the $CH_2O$ column above the P-3B is on the order of $2 \times 10^{15}$ molecules cm$^{-2}$ from 4 to 14 September, and $1 \times 10^{15}$ molecules cm$^{-2}$ from 24 to 26 September flights. Free tropospheric $CH_2O$ in the model is much larger than that observed by the in situ instrument during several early flights, possibly due to the transport of too much boundary layer air in the model (Fried et al., 2016b). We find its removal introduces daily background biases that reduce the overall correlation between P-3B and GCAS observations; as a result, we do not remove the modeled $CH_2O$ above the spiral from the GCAS results in these comparisons.

### 6.1.1 NO₂

The overall correlation between the P-3B P-CL and GCAS $NO_2$ measurements is very good ($r^2 = 0.89$). The two instruments also agree well in magnitude, with GCAS slightly lower than the P-3B at larger $NO_2$ columns by ~10 %. At background levels, GCAS overestimates the P-3B columns by ~$1.6 \times 10^{15}$ molecules cm$^{-2}$. This background offset is most likely due to a combination of uncertainties introduced by the GCAS stratospheric correction and the modeled tropospheric background



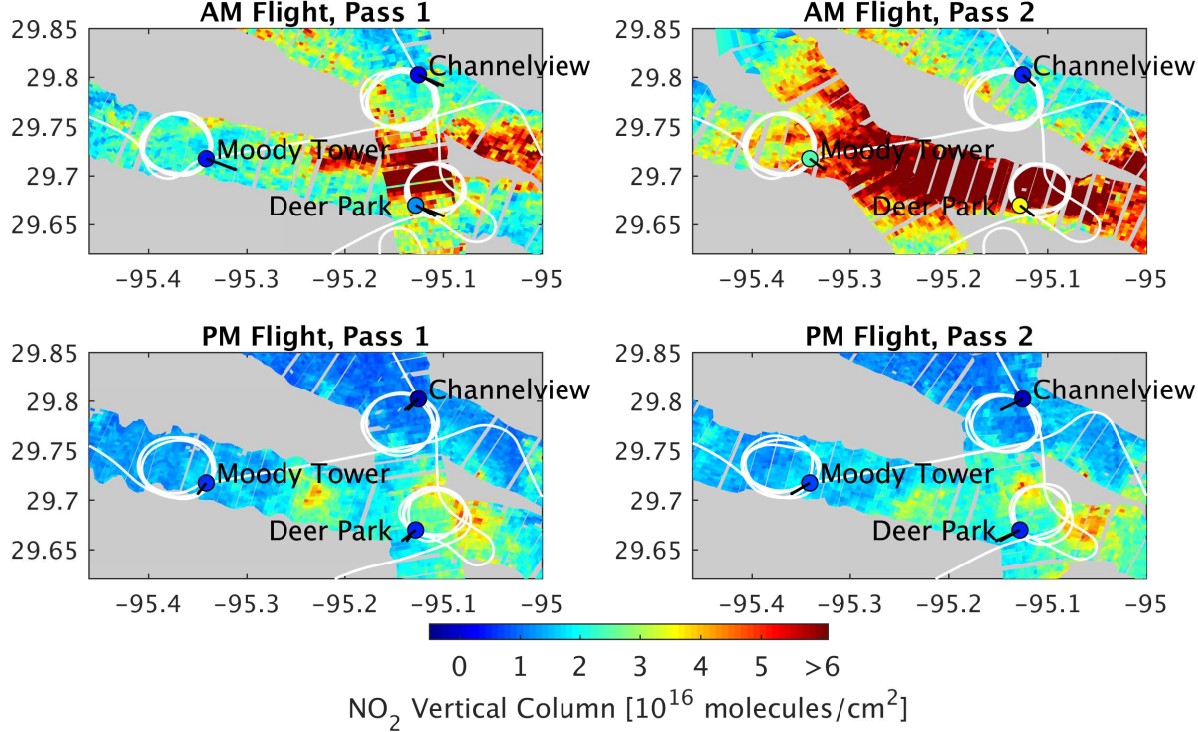

**Figure 5.** GCAS tropospheric $NO_2$ columns measured near DISCOVER-AQ ground sites in the area of downtown Houston on 25 September 2013. P-3B flight tracks are shown in white. Pandora direct sun $NO_2$ tropospheric columns (total column $NO_2$ minus modeled $NO_2$ above the aircraft) are shown in filled circles. Black lines represent the line of sight of each Pandora in intersecting the bottom 2 km of the atmosphere. The largest $NO_2$ column observed by GCAS on this day was $16 \times 10^{16}$ molecules cm$^{-2}$. The $NO_2$ precision at this resolution is ~$1 \times 10^{15}$ molecules cm$^{-2}$.

column in the reference spectrum in Equation 4, with a possible contribution from the uncertainty in the column below the minimum P-3B spiral altitude. Most of the variability observed in the $NO_2$ column comparisons is due to the large radius of the P-3B spiral, which can mean the P-3B flies in and out of $NO_2$ plumes in some spirals, as is shown in Fig. 5, as well as the inability of the P-3B to capture profiles of near-surface $NO_2$ below 300 m near the Channelview, Manvel Croix and Moody

5  Tower sites.

### 6.1.2  $CH_2O$

The agreement between the P-3B DFGAS and GCAS $CH_2O$ columns is also reasonably good ($r^2 = 0.54$), with GCAS on average 8 % larger than DFGAS. There appears to be little background offset bias influence from the reference spectrum, although the GCAS columns are likely overestimated by some small amount as the $CH_2O$ above the P-3B has not been

10  removed, as discussed previously. The $CH_2O$ spatial features in our GCAS observations are often more diffuse than those of




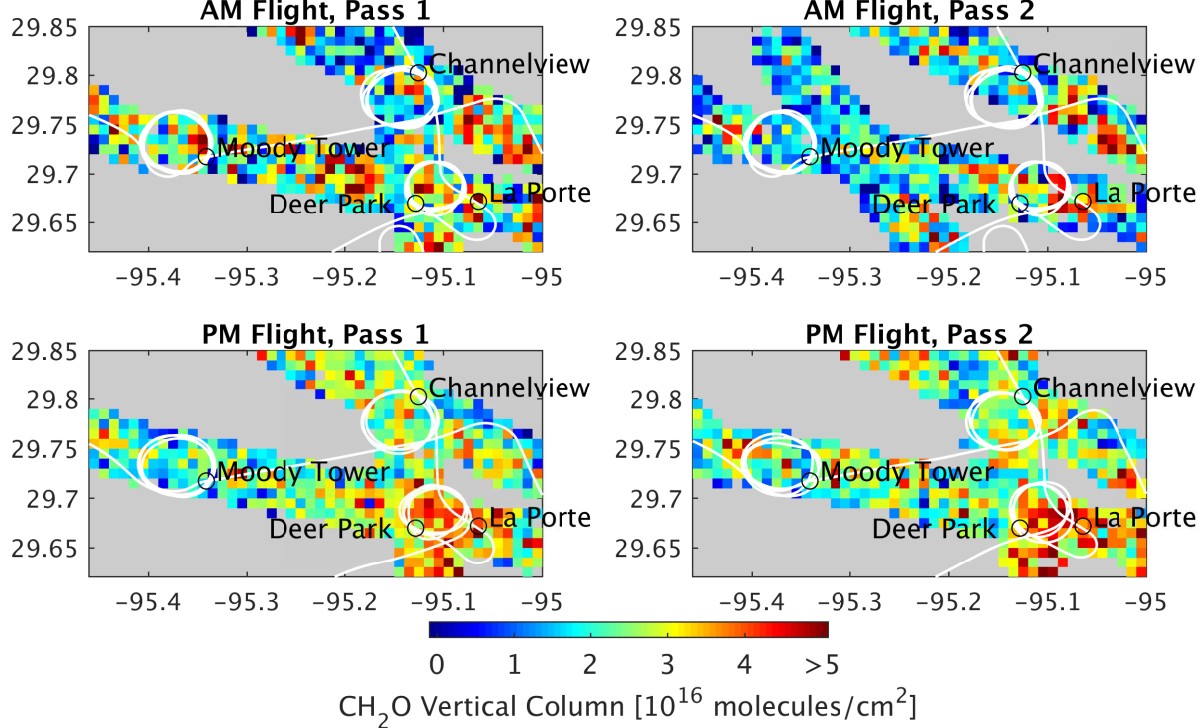

**Figure 6.** GCAS tropospheric $CH_2O$ columns measured near DISCOVER-AQ ground sites in the area of downtown Houston on 25 September 2013. P-3B flight tracks are shown in white. $CH_2O$ columns are spatially averaged on a $0.01° \times 0.01°$ grid (~1 $km^2$). The $CH_2O$ precision at this resolution is ~$1 \times 10^{16}$ molecules $cm^{-2}$.

$NO_2$, and modeled profile shapes during the spirals are often in close agreement with the P-3B observations in the boundary layer, indicating that the profile shapes used in the GCAS $CH_2O$ AMF calculations are likely reliable in the boundary layer overall. Large columns are often seen at Deer Park and Channelview near industrial facilities, and at Conroe and West Houston (likely from biogenic sources as well as transport from the industrial regions).

### 5  6.1.3  AMF from P-3B profiles

In order to assess the dependence of the GCAS observations on the profile uncertainty, we also apply the P-3B profiles in place of model profiles in the GCAS AMF calculations. In this case, when the spiral profile is applied to the GCAS observations within its vicinity, the $NO_2$ correlation remains the same but increases to $r^2 = 0.62$ for $CH_2O$. On average, the use of the observed profiles lowers the GCAS tropospheric $NO_2$ and $CH_2O$ column estimates by a few percent. In the case of $NO_2$, the reduced major axis linear regression of GCAS as a function of P-3B column results in a change in the slope from 0.90 to 0.86 and in the intercept from $1.7 \times 10^{15}$ molecules $cm^{-2}$ to $1.6 \times 10^{15}$ molecules $cm^{-2}$. For $CH_2O$, the slope changes from 1.08 to 1.06, and the intercept from $-5 \times 10^{14}$ molecules $cm^{-2}$ to $3 \times 10^{14}$ molecules $cm^{-2}$.





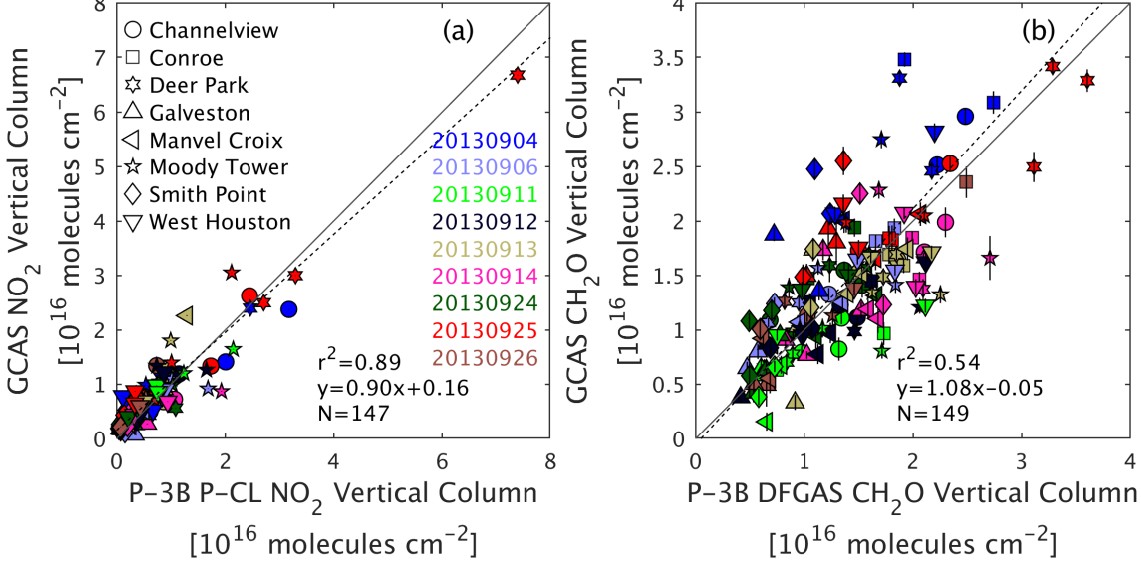

**Figure 7.** Columns derived from in situ measurements of (a) $NO_2$ from the chemiluminescence instrument and (b) $CH_2O$ from the DFGAS instrument on the P-3B aircraft compared with vertical columns measured by the GCAS instrument, over nine days during the DISCOVER-AQ Texas campaign. Each GCAS vertical column is the mean of all retrieved cloud-free GCAS columns below the aircraft within 5 km and 1 h of its coincident P-3B spiral center. GCAS air mass factors are determined using modeled CMAQ profiles. The solid line represents the 1:1 ratio. The dotted line represents the reduced major axis linear regression.

## 6.2 Pandora $NO_2$ column measurements

Figures 8 and 9 show comparisons of GCAS tropospheric columns with Pandora $NO_2$ columns derived from both direct sun and MAX-DOAS scattered light retrievals, by day and by site. Figure 10 shows the Pandora measurements at four sites as a function of time, and GCAS coincidences with those observations. In the case of the direct sun Pandora observations, we

5   have estimated the tropospheric Pandora column by subtracting the modeled $NO_2$ above the GCAS instrument (typically ~2.5–4×$10^{15}$ molecules cm$^{-2}$) from the closest Pandora observation in time within 3 minutes. In the case of MAX-DOAS comparisons, we compare a single GCAS observation over each site with the closest MAX-DOAS observation within 20 min. For the comparison with direct sun observations, we have determined the GCAS observation from the mean of GCAS ground pixels intersected by the Pandora line-of-sight in the bottom 2 km of the atmosphere (shown in Fig. 5). This helps to

10   minimize the influence of the Pandora viewing geometry on the comparison. For instance, GCAS consistently measures large columns over the Deer Park site, with some of the largest $NO_2$ often to the north of the site; however, when viewing the sun directly, Pandora always looks south into cleaner air. The use of a GCAS $NO_2$ amount determined along the Pandora direct sun line-of-site reduces the influence of these biased site locations on the results, with an overall reduction in the GCAS versus





Pandora bias of 20 %. There remain, however, several sites with an obvious difference in GCAS versus Pandora direct sun measurements, despite considering the field-of-view.

Overall, GCAS tropospheric $NO_2$ is larger than Pandora (GCAS/Pandora=1.50 for direct sun and 1.33 for MAX-DOAS), although the spatial correlations are very good at $r^2 = 0.85$ (direct sun) and $r^2 = 0.94$ (MAX-DOAS). A background offset
of ~$2 \times 10^{15}$ molecules $cm^{-2}$ is seen between GCAS and the Pandora direct sun measurements, similar to that seen in the P-3B comparisons. Again, this is most likely from uncertainties in the modeled stratospheric correction and reference spectrum correction, with a possible contribution from the Pandora reference as well. More surprisingly, GCAS $NO_2$ is consistently larger than Pandora measurements at high $NO_2$ values. The larger GCAS values could be influenced by several uncertainties that can result in cumulative biases in the AMF calculation, caused by potential uncertainties in surface reflectance, profile
shape and aerosols.

Previous airborne comparisons with the GeoTASO instrument during DISCOVER-AQ Texas on four relatively unpolluted or cloudy days (13, 14, 18, 24 September) also suggested airborne $NO_2$ larger than Pandora (Nowlan et al., 2016). Souri et al. (2018) also found a large difference between GCAS and Pandora observations during the Texas campaign. By using a Bayesian inversion to constrain the MODIS BRDF, they reduced the overestimation of GCAS relative to Pandora by 23
% through a 0.023 increase in surface albedo, broadly consistent with studies that have found a low bias in MODIS surface reflectance (Wang et al., 2010b; Salomon et al., 2006) at short wavelengths. The exclusion of aerosols in our AMF calculation may cause the AMF to be underestimated (and therefore the vertical column to be overestimated) in some cases, particularly where scattering aerosols are in the lowest part of the boundary layer (see discussion in Section 4.7.2). We find that the GCAS vertical columns at Pandora coincidences are reduced on average by 10 % when the air mass factor is calculated using the
nearest HSRL aerosol optical thickness profiles below the aircraft for scattering aerosols. In the previous section, we saw that P-3B observations point to a small +4 % bias in the GCAS column from the use of CMAQ modeled $NO_2$ profiles, on average.

Differences in the GCAS and Pandora slant column retrievals themselves may also play a role, including the wavelength fitting region and atmospheric temperature assumptions. The Pandora slant column product used in our study was produced assuming a fixed effective temperature of 264 K, which could result in a low bias in the retrieved Pandora slant column of 10
% (Spinei et al., 2014). Previous comparisons of Pandora direct sun total column $NO_2$ observations with other ground based observations have shown good agreement (Herman et al., 2009; Wang et al., 2010a), while Knepp et al. (2017) compared a year of retrieved Pandora zenith-sky stratospheric $NO_2$ slant columns with those from a Network for the Detection of Atmospheric Composition Change (NDACC) spectrometer, using different retrieval settings, and found Pandora underestimated the NDACC instrument by 7–40 %, with the bias dependent on season and solar zenith angle.

Different factors likely dominate the uncertainties at different sites; some sites are located at locations with very inhomogeneous surface reflectance (Smith Point and Moody Tower) and some at locations with large uncertainties in profile shape. The slope is also dominated by the larger polluted measurements on the 25 September, which was a day with complicated meteorology, a morning boundary layer of ~200 m (according to HSRL data), and uncertain emissions (Souri et al., 2018). Despite the sources of uncertainty on the GCAS columns, it should be noted that a large reduction in the GCAS vertical columns





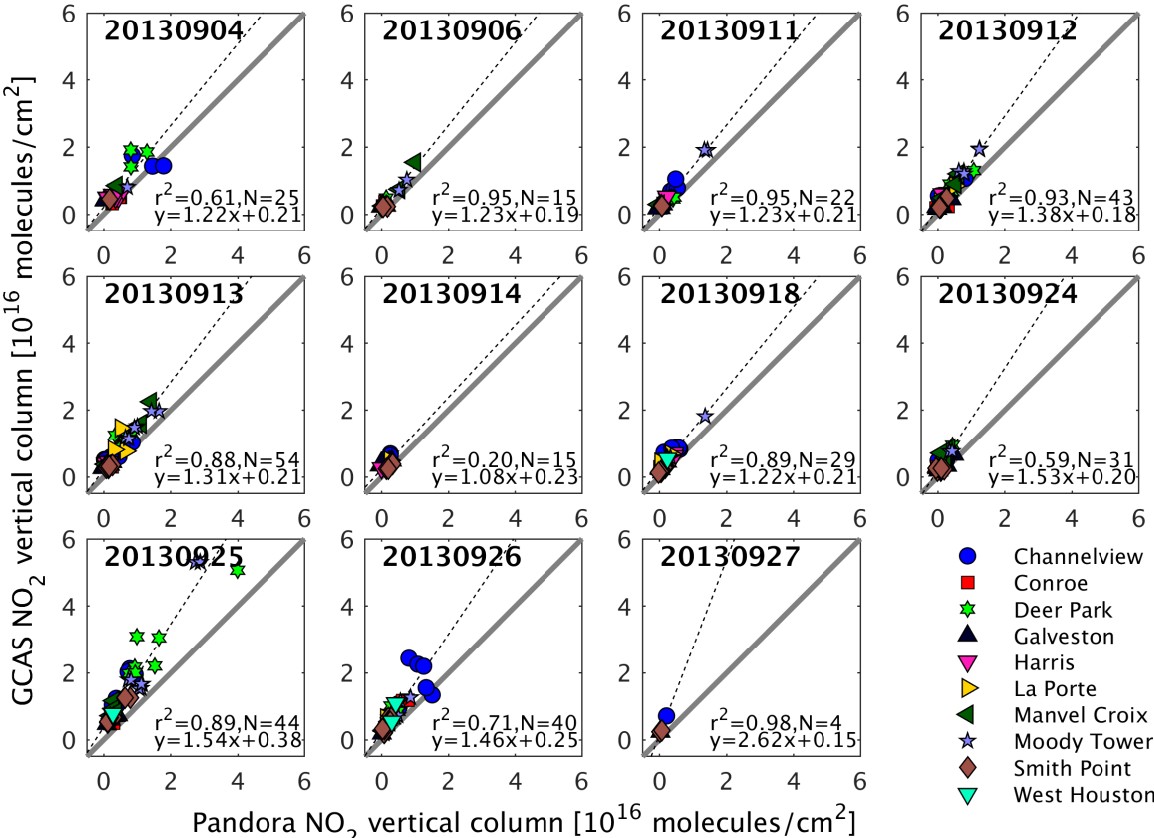

**Figure 8.** Pandora direct sun $NO_2$ tropospheric columns vs. GCAS $NO_2$ tropospheric columns by day for cloud-free observations over Houston during DISCOVER-AQ Texas 2013. The Pandora columns are the total $NO_2$ columns measured by Pandora minus the colocated modeled stratospheric $NO_2$ columns used in the GCAS analysis. All correlations are statistically significant at the $p < 0.001$ except for those of 14 ($p = 0.09$) and 27 ($p = 0.01$) September. The solid line represents the 1:1 ratio. The dotted line represents the reduced major axis linear regression.

from the use of different AMF inputs that resulted in better agreement with the Pandora columns could mean a significant underestimation of both the $NO_2$ and $CH_2O$ P-3B columns.

# 7 Conclusions

We have presented trace gas retrievals of $NO_2$ and $CH_2O$ from the GCAS instrument during the DISCOVER-AQ Texas 2013
5 campaign. In these retrievals, we first use a spectral fit to derive slant column densities from nadir spectra, in combination with reference spectra measured over a clean area. We then convert those slant columns to vertical columns using tropospheric



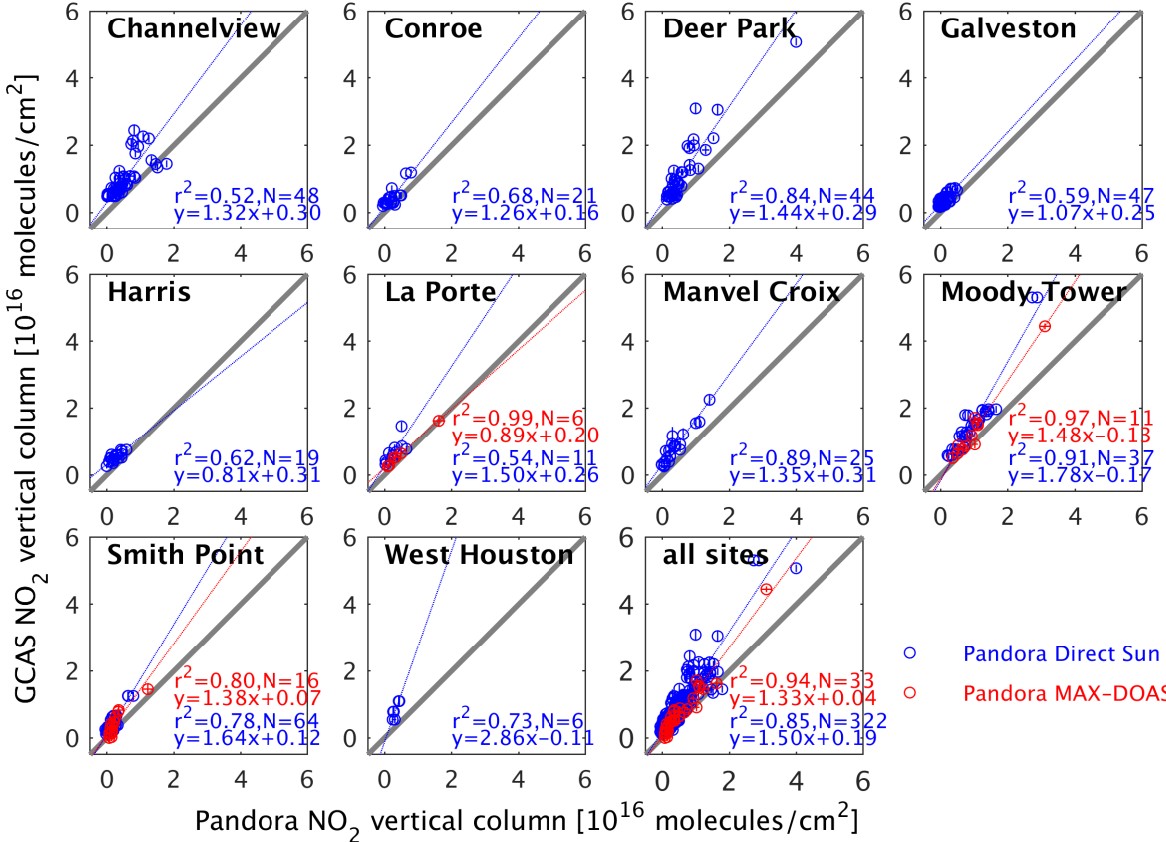

**Figure 9.** Pandora $NO_2$ tropospheric columns from direct sun and MAX-DOAS observations vs. GCAS $NO_2$ tropospheric columns by site for cloud-free observations over Houston during DISCOVER-AQ Texas 2013. The Pandora direct sun columns are the total $NO_2$ columns measured by Pandora minus the co-located modeled stratospheric $NO_2$ columns used in the GCAS analysis. The solid line represents the 1:1 ratio. The dotted lines represent the reduced major axis linear regressions.

trace gas profiles from the CMAQ model and surface reflectance from the MODIS BRDF product. At a spatial resolution of $250\,\mathrm{m} \times 500\,\mathrm{m}$, the $NO_2$ product has a mean precision of $1 \times 10^{15}$ molecules $\mathrm{cm}^{-2}$, and the $CH_2O$ product has a mean precision of $1.9 \times 10^{16}$ molecules $\mathrm{cm}^{-2}$. In order to meet TEMPO precision requirements, and to detect enhanced $CH_2O$ during the DISCOVER-AQ Texas campaign, we recommend $CH_2O$ be spatially averaged to $1\ \mathrm{km}^2$. Uncertainties in $NO_2$

5 polluted observations are dominated by air mass factor uncertainties, which result primarily from uncertainties in surface reflectance, aerosol loading and trace gas profile shape. These air mass factor uncertainties also play a role in individual $CH_2O$ uncertainties, but can be similar in magnitude to uncertainties from spectral fitting noise.

Comparisons between GCAS and P-3B and Pandora observations show GCAS data are very well correlated with these coincident measurements, but in some cases show differences in magnitude. GCAS columns agree well with those inferred





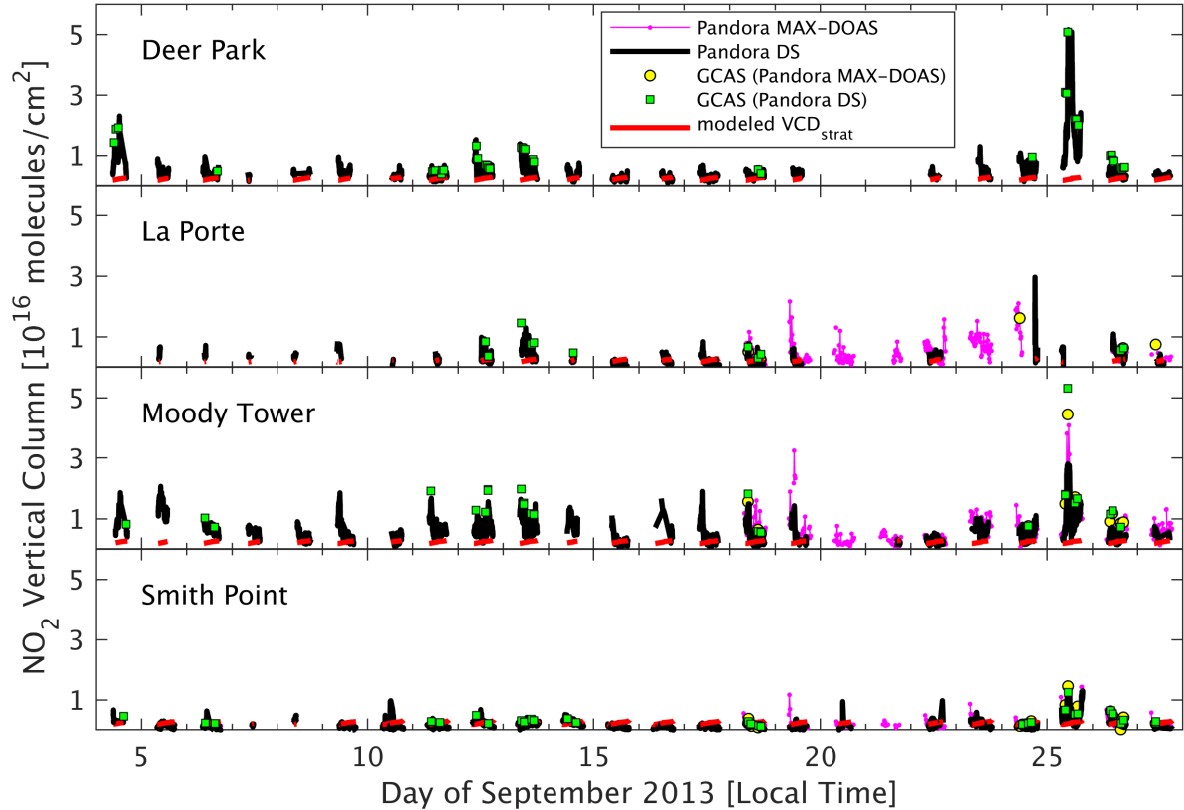

**Figure 10.** Tropospheric $NO_2$ columns from Pandora direct sun (DS) and MAX-DOAS observations as a function of time between 4 and 27 September at Deer Park, La Porte, Moody Tower and Smith Point sites, and GCAS coincidences with those observations, as well as stratospheric $NO_2$ from a model at Pandora direct sun measurements. Pandora direct sun tropospheric columns are derived by removing the modeled stratosphere from the retrieved Pandora total columns.

from P-3B in situ profiles for both $NO_2$ ($r^2 = 0.89$, GCAS/P-3B slope=0.90 and intercept=$1.6 \times 10^{15}$ molecules cm$^{-2}$) and $CH_2O$ ($r^2 = 0.54$, GCAS/P-3B slope=1.08 and intercept=$-5 \times 10^{14}$ molecules cm$^{-2}$). The use of P-3B profiles in GCAS air mass factor calculations indicates a mean uncertainty of 2-4 % in GCAS columns from the use of a modeled profile shape over Houston. GCAS is higher than Pandora MAX-DOAS tropospheric $NO_2$ columns but shows excellent spatial agreement

5  ($r^2 = 0.94$, GCAS/Pandora slope=1.33 and intercept=$4 \times 10^{14}$ molecules cm$^{-2}$); these differences in magnitude, however, remain within the bounds of GCAS systematic error estimates in the AMF. The largest discrepancies in magnitude are seen between GCAS and Pandora direct sun observations, although spatial correlations are very good ($r^2 = 0.85$, GCAS/Pandora slope=1.50 and intercept=$1.9 \times 10^{15}$ molecules cm$^{-2}$). As both Pandora and GCAS are key instruments in planned TEMPO





validation activities, there is clearly a need to resolve these differences in magnitude to ensure reliable validation studies. Further opportunities for comparisons over different geographic areas and pollution regimes exist in other campaigns.

Since DISCOVER-AQ Texas in 2013, the airborne GCAS and GeoTASO instruments have been deployed in the DISCOVER-AQ Colorado field campaign (2014), KORUS-AQ field campaign (2016), GOES-R Validation Campaign (2017) and Lake

5   Michigan Ozone Study (2017). These data are currently under study, and offer further opportunities to examine the effects of surface characterization, profile shape, aerosols, viewing geometries and trace gas heterogeneity on ground, airborne and satellite remotely-sensed trace gas columns.

## 8   Data availability

The GCAS and P-3B $NO_2$ and $C_2HO$ data and Pandora direct sun $NO_2$ columns are publicly available from the DISCOVER-

10   AQ data archive at http://www-air.larc.nasa.gov/missions/discover-aq/discover-aq.html (doi: 10.5067/Aircraft/DISCOVER-AQ/Aerosol-TraceGas). The archived GCAS data also include coincident model profiles for each observation.

*Acknowledgements.* This study was supported under NASA grants NNX14AR69G and NNX17AE09G. MODIS MCD43GF V005 data were provided by the MODIS remote sensing group at the University of Massachusetts Boston. We thank Amir Souri for helpful discussions.





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



**Table 1.** Summary of GCAS flights during DISCOVER-AQ Texas 2013. Times are Local Time (LT) (UTC - 5 hours). Days with P-3B aircraft flights are denoted by an X in the rightmost column.

| Date | Description | Flight Time (AM) | Flight Time (PM) | P-3B ? |
|---|---|---|---|---|
| 4 September | Houston | 08:46–12:05 | 13:37–17:12 | X |
| 6 September | Houston | 08:47–12:04 | 13:59–17:13 | X |
| 10 September | Ocean color | 07:57-10:10 | 15:03–17:19 | |
| | | 11:41–12:59 | | |
| 11 September | Houston | 08:47–12:06 | 13:39–16:50 | X |
| 12 September | Houston | 08:47–10:44 | 13:42–17:00 | X |
| 13 September | Houston | 08:41–12:14 | 13:56–17:17 | X |
| 14 September | Houston | 07:53–11:23 | 12:26–15:52 | X |
| 17 September | Ocean color | 07:55–11:11 | 13:45–17:09 | |
| 18 September | Houston | 08:43–12:16 | 14:06–17:31 | |
| 24 September | Houston | 08:42–12:00 | 13:12–16:25 | X |
| 25 September | Houston | 08:45–12:02 | 13:50–17:10 | X |
| 26 September | Houston | 08:40–11:50 | 14:18–17:41 | X |
| 27 September | Houston | 08:39–12:04 | | |

**Table 2.** DISCOVER-AQ sites with Pandora spectrometers overflown by GCAS. The Pandora ID is a identification number given to each individual Pandora instrument. Asterisks indicate Pandoras used for MAX-DOAS measurements; all other Pandoras were used solely for direct sun measurements. The mean GCAS overpass time of Pandora sites is 10:07 LT (earliest/latest 08:18/11:51 LT) for morning flights and 15:25 LT (earliest/latest 12:51/17:12 LT) for afternoon flights.

| Site | Latitude (°) | Longitude (°) | Pandora ID |
|---|---|---|---|
| Channelview | 29.803 | −95.126 | P26 |
| Conroe | 30.350 | −95.425 | P31 |
| Deer Park | 29.670 | −95.128 | P32 |
| Galveston | 29.254 | −94.861 | P34 |
| Northwest Harris County | 30.039 | −95.674 | P30 |
| La Porte | 29.672 | −95.065 | P38*, P39 |
| Manvel Croix | 29.520 | −95.392 | P33 |
| Moody Tower | 29.718 | −95.341 | P28, P35* |
| Smith Point | 29.546 | −94.787 | P8, P29*, P36 |
| West Houston | 29.833 | −95.657 | P18 |



**Table 3.** Fitting details and fitted parameters used in GCAS trace gas retrievals.

| Parameter | NO$_2$ Retrieval | CH$_2$O Retrieval |
| --- | --- | --- |
| Fitting window | 420.0–465.0 nm | 328.5–356.5 nm |
| NO$_2$ cross section | Vandaele et al. (1998), 294 K | Vandaele et al. (1998), 294 K |
| CH$_2$O cross section | N/A | Chance and Orphal (2011), 300 K |
| O$_3$ cross section | Brion et al. (1993), 218 and 295 K | Brion et al. (1993), 218 and 295 K |
| H$_2$O vapor cross section | Rothman et al. (2013), 288 K, 1 atm | N/A |
| BrO cross section | N/A | Wilmouth et al. (1999), 228 K |
| O$_2$–O$_2$ cross section | Thalman and Volkamer (2013), 293 K | Thalman and Volkamer (2013), 293 K |
| Undersampling | Chance et al. (2005) | Chance et al. (2005) |
| Ring spectrum | Chance and Spurr (1997) | Chance and Spurr (1997) |
| Scaling polynomial | 5th order | 5th order |
| Baseline polynomial | 4th order | 4th order |
| Wavelength shift | | |