# Peer review of "Nitrogen dioxide and formaldehyde measurements from the GEOstationary Coastal and Air Pollution Events (GEO-CAPE) Airborne Simulator over Houston, Texas"

_Atmospheric Measurement Techniques, 2018_

## Referee Comment (RC1) · Anonymous Referee #2 · 8 Aug 2018

The paper reports results of the first intensive field campaign (Discover AQ, Texas 2013) of GCAS, a compact 2-channel airborne spectrometer. GCAS NO2 and HCHO retrievals are compared with trace gas columns derived from coincident in situ profile measurements made by instruments on a P-3B aircraft, and with NO2 observations from ground-based Pandora spectrometers, operating in direct sun and scattered light modes. In a previous paper, Nowlan et al. (2016), preliminary GCAS retrievals were compared with GEOTASO retrievals.

The paper provides a detailed overview of all retrieval steps and comparison with co-

incident measurements, including uncertainties, reasons for observed differences, etc. The paper contains useful content for the preparation of validation campaigns for the new generation of air quality sensors, such as TEMPO. These campaigns will likely involve GEOTASO, GCAS and profile flights for the airborne segment and Pandora spectrometers for the ground-based segment. The paper fits well within the scope of AMT, is well-written and well-structured. However, some revisions need to be conducted in the paper before publication.

General comments:

-What is a bit missing in the paper is a geophysical interpretation of the acquired data. Typically there were four similar flights per day over the area which provides a good view on the spatial distribution of NO2 and HCHO. Even if it is beyond the scope of this paper, a section discussing the changes in the trace gas field and possible explanations would be an added-value to the paper. There are a few sentences in Section 5 on this, but it could be more extended.

-Secondly, it is not clear to me why profile shapes from CMAQ model output at 4 by 4 km are used for the AMF calculations, while you have NO2 and HCHO in-situ measurements from P-3B spiral flights available. I can understand this approach for the comparison with P-3B (Sect. 6.1) as you need independent data, but it is not clear why you don't use the P-3B profile shapes instead of CMAQ when comparing to PANDORA spectrometers (Sect. 6.2). I would expect that the CMAQ model does not represent the strong spatial variability of the 3D NO2 field, which you can expect in an urban area, something you mention as well in the manuscript (p.14, L.20). On the other hand, the dependency on the profile uncertainty was assessed in 6.1.3, and seems to be small.

Specific comments:

P3, L10: Please add: Tack, F., Merlaud, A., Meier, A. C., Vlemmix, T., Ruhtz, T., Iordache, M.-D., Ge, X., van der Wal, L., Schuettemeyer, D., Ardelean, M., Calcan, A., Schönhardt, A., Meuleman, K., Richter, A., and Van Roozendael, M.: Intercomparison of four airborne imaging DOAS systems for tropospheric NO2 mapping – The AROMAPEX campaign, Atmos. Meas. Tech. Discuss., https://doi.org/10.5194/amt-2017-478, in review, 2018.

P4, L1: Please mention also explicitly the swath width at this altitude.

P4, L12: Explain shortly the impact of not having a zenith sky reference measurement capability when compared to GEOTASO or ACAM.

P4, L23: Maybe mention here already that GeoTASO and GCAS were intercompared in the previous paper.

P.5, L1: It would be interesting to mention the main drivers for the chosen flight path in section 3.1. Driven by Sources? Pandora sites? . . .

P.7, L.8: Please specify the vertical resolution (for the part below the aircraft).

P.8, L.10: Note that the spectral performance can be affected by in-flight pressure changes as well: See Kuhlmann, G., Hueni, A., Damm, A., and Brunner, D.: An Algorithm for In-Flight Spectral Calibration of Imaging Spectrometers, Remote Sensing, 8, 1017, doi:10.3390/rs8121017, 2016.

P.9, L13: Maybe I missed it but I could not find back the estimated reference spectrum amount. The estimation also affects the VCD uncertainty and should be mentioned in Section 4.7. You mention it as an uncertainty on P. 21, L6.

P.9, L30: Why is the fitting window 420-465 nm for NO2 and for example not 425-490 nm as proposed by NDACC (for MAX-DOAS instruments)? Please clarify. On p. 3, L27 the wavelength range of the UV/Visible channel is 300-490 nm.

P.11, L8: Please mention the spatial resolution of the MODIS BRDF product.

P.13, L.5: Please specify the impact of the spatial binning on the uncertainty for HCHO.

P. 13, L.32: The aerosol effect is also well explained in Meier, A. C., Schönhardt, A.,

Bösch, T., Richter, A., Seyler, A., Ruhtz, T., Constantin, D.-E., Shaiganfar, R., Wagner, T., Merlaud, A., Van Roozendael, M., Belegante, L., Nicolae, D., Georgescu, L., and Burrows, J. P.: High-resolution airborne imaging DOAS measurements of NO2 above Bucharest during AROMAT, Atmos. Meas. Tech., 10, 1831-1857, https://doi.org/10.5194/amt-10-1831-2017, 2017.

P.14, L.12: Were there no aerosol instruments on the P-3B aircraft? In case not, it would be a real added-value for future campaigns, as you are already doing the efforts to perform spiral flights for NO2 and HCHO profiles. A good knowledge of the AOD profiles can indeed drastically decrease the uncertainties on the AMF.

P.15,L.27: Specify "near-surface" in order to avoid confusion with surface concentrations.

P.16, Figure3: Indication of the average wind direction (and speed) could help interpretation or describe in text if too variable (same for the other maps).

P.18, Figure 5: Please explain the reason for missing data. Sometimes a full across-track scanline is missing which is not related to cloud filtering. Are there other filters applied?

P.20, Figure 7b: Error bars not explained in text?

P.21, L.6: What about the use of the simple geometric AMF for PANDORA (P.6, L3). Could this also contribute (significantly) to the differences observed?

Technical corrections:

P3, L1-L7: This paragraph is a bit too technical and would be better fitting in section 2.

P.15, L.13: Change "individual industrial sites" to "single industrial sites" or "single stacks".

P.20, L13: Change line-of-site to line-of-sight

[Figure]

P.21, L27: Please reformulate as sentence is not clear. Maybe: ". . ..with retrievals from a spectrometer from the Network for the Detection of Atmospheric Composition Change (NDACC) using. . ." + Please specify also the type of (DOAS) instrument.

P.22, L1: Please put comma after "inputs" and after "columns".

P.24, Figure10: acronym DS is used here for the first time. Should also be used in text and written in full at first appearance.

---

## Referee Comment (RC2) · Anonymous Referee #1 · 23 Aug 2018

**1  General comments**

The paper describes the retrieval of NO2 and CH2O columns and their uncertainties from GCAS instrument during DISCOVER-AQ campaign and compares them with in-situ aircraft and ground-based profile and column observations. The topic of the paper is well within the scope of AMT.

The paper is mostly well structure, but discussions of the results are currently spread

over various sections. I suggest adding a "discussions" section at the end of the paper to discuss the large biases between GCAS, Pandora and P-3B in context of the expected uncertainties of the three different datasets.

The description of the retrieval uncertainties is very detailed but does not include uncertainties form the reference columns. Furthermore, the total expected uncertainty of GCAS NO2 and CH2O columns due to instrument noise, AMF uncertainties and reference column is not calculated but is important to understand if the results of the comparison are within the expected uncertainties.

Furthermore, a calculation of the expected uncertainties from the preparation of the Pandora and P-3B data for comparison with the GCAS columns is also missing. In particular, the effects of cutting and extending the columns using the (uncertain) model data should be calculated.

Finally, the description of the vertical column results (Section 5) is difficult to understand, because the locations of the discussed landmarks are not marked in figures 3 to 6. It would be beneficial to update the figures or provide larger version in the supplement.

In summary, I recommend publication of this paper in AMT after minor revision.

**2  Specific comments**

P8, L4: rename section from "Calibration" to "Spectral calibration".

P8, L12-14: The sentence on radiometric calibration can be removed, because it seems not to be relevant to the paper.

P8, L29: Please add a paragraph about the results of the spectral calibration (wavelength shifts, slit function and their uncertainties) and the impact on the SCD uncertainty.

P8-9: Sections 4.3 and 4.4 could be subsections of 4.2; similar to the subsections of section 4.5

P9, L28ff: Section 4.4 would fit better at the end of section 4.2

P10, L9: The term "differential slant column" should be already introduced in Section 4.2, because it makes the usage of the reference spectrum clearer.

P11, L8ff: Please state temporal and spatial resolution of the BRDF product.

P11, L14: Please give a number for "a few percent".

P12, L4ff: How large is the SCD uncertainty due to uncertainties in the spectral calibration?

P18, L10: Replace "more diffuse" with "nosier".

P19, L9: Please give a number for "a few percent".

P20, Figure 7: The figure is very crowded and error bars are not explained for GCAS and missing for P-3B (extrapolation errors?).

P21, L 8-10: The sentence is very vague. Could you provide numbers?

**3 Technical corrections**

P10, L5: maybe replace "absorber" with "trace gas"

P10, L7: add a "," after "atmosphere"

---

## Author Response (AR1)

**Response to Referee #1**

We thank Referee #1 for helpful comments. We have made several changes to our manuscript based on these suggestions. In the following text, the referee's comments are in bold italics, followed by our response. The manuscript with changes tracked follows this response.

**1 General comments**

The paper describes the retrieval of NO2 and CH2O columns and their uncertainties from GCAS instrument during DISCOVER-AQ campaign and compares them with insitu aircraft and ground-based profile and column observations. The topic of the paper is well within the scope of AMT.

The paper is mostly well structure, but discussions of the results are currently spread over various sections. I suggest adding a "discussions" section at the end of the paper to discuss the large biases between GCAS, Pandora and P-3B in context of the expected uncertainties of the three different datasets.

We have added a section at the end of Section 6 discussing these results. Some earlier text has been moved to this section, and some added for clarity. Please see manuscript for changes to the text.

**The description of the retrieval uncertainties is very detailed but does not include uncertainties form the reference columns. Furthermore, the total expected uncertainty of GCAS NO2 and CH2O columns due to instrument noise, AMF uncertainties and reference column is not calculated but is important to understand if the results of the comparison are within the expected uncertainties.**

We have added a section (Section 4.5.3) on uncertainties from modeled columns used to correct for the reference and model the stratosphere, and have added a section on the total uncertainty (Section 4.5.4).

**Furthermore, a calculation of the expected uncertainties from the preparation of the Pandora and P-3B data for comparison with the GCAS columns is also missing. In particular, the effects of cutting and extending the columns using the (uncertain) model data should be calculated.**

We have rearranged the P-3B section to discuss the preparation of the columns in its own subsection. We have added the following text to describe uncertainties from the column calculations of the P-3B: "A large source of error from these extrapolations is the inhomogeneity of the trace gas field, which is particularly strong for NO2 (see Fig. 5 for example), as the lowest mixing ratio could be measured in or out of an area of high NO2, and is then extended to the ground. Lamsal et al. (2014) estimated errors in the DISCOVER-AQ Maryland P-3B NO2 columns of generally less than 20 % from extrapolation of the NO2 profile below ~300 m, assuming a factor of two error in the extrapolation."

"As CH2O gradients near the surface tend to be smaller than those of NO2, the extrapolation error is also likely less significant. P-3B CH2O columns calculated with an extrapolated model gradient and a direct extrapolation vary by about 5 %."

"Comparisons of CMAQ and P-3B NO2 profiles in the free troposphere (3–5 km) suggest a mean absolute error of 70 % in the free troposphere (CMAQ is 10 % higher than the P-3B on average). If we assume similar discrepancies above the highest P-3B altitude, this may lead to an uncertainty of ~2 x  $10^{14}$  molecules cm-2 in the GCAS column from the removal of the column above the P-3B."

"The mean absolute error from CMAQ versus P-3B between 2–5 km is 40 %, with much larger biases of ~100 % on certain days. We find its removal introduces daily background biases that reduce the overall correlation between P-3B and GCAS observations; as a result, we do not remove the modeled CH2O above the spiral from the GCAS results in these comparisons. This results in an uncertainty on the order of 1-3 x  $10^{15}$  molecules cm-2, depending on the flight."

We have added the following to the Pandora section: "The uncertainty in the stratospheric NO2 column in our model is estimated at 30 % (see Section 4.5.3).".

**Finally, the description of the vertical column results (Section 5) is difficult to understand, because the locations of the discussed landmarks are not marked in figures 3 to 6. It would be beneficial to update the figures or provide larger version in the supplement. In summary, I recommend publication of this paper in AMT after minor revision.**

We have added landmarks and roads to the summary Figure 1. We have also added the caption "Major roads are shown in yellow. ExxonMobil Baytown and Texas City are large petrochemical and petroleum refinery complexes. The Baytown complex lies near the entrance to the main part of the Houston Ship Channel industrial area, which ends 6.5 km to the east of the downtown."

**2 Specific comments**

*P8, L4: rename section from "Calibration" to "Spectral calibration".* Changed in text.

**P8, L12-14: The sentence on radiometric calibration can be removed, because it seems not to be relevant to the paper.**

We have removed this sentence.

**P8, L29: Please add a paragraph about the results of the spectral calibration (wavelength shifts, slit function and their uncertainties) and the impact on the SCD uncertainty.**

We have added a paragraph discussing the results. The impact on SCD is discussed in a later section.

**P8-9:* Sections 4.3 and 4.4 could be subsections of 4.2; similar to the subsections of section 4.5**

**P9, L28ff: Section 4.4 would fit better at the end of section 4.2**

We have moved Sections 4.3 and 4.4 to subsections of 4.2 as suggested, and have changed the first paragraph of Section 4.2 to make it a subsection ("Spectral fitting") for clarity.

**P10, L9: The term "differential slant column" should be already introduced in Section 4.2, because it makes the usage of the reference spectrum clearer.**

We have modified the sentence slightly and now introduce the definition of the differential slant column: "The retrieved differential slant columns are represented by  $x_i$ . These differential slant columns are the differences between the slant columns in the nadir observation of interest and the slant columns in the reference spectrum."

**P11, L8ff: Please state temporal and spatial resolution of the BRDF product.**

We have added the following to the text: "This BRDF product is provided at a spatial resolution of 30 arcsec (~0.80 km in longitude by 0.92 km in latitude over Houston) every 8 days, based on 16 days of MODIS measurements."

**P11, L14: Please give a number for "a few percent".**

We have modified this sentence to read: "These results are typically within 2–3 % of the results derived ..."

**P12, L4ff: How large is the SCD uncertainty due to uncertainties in the spectral calibration?**

We have added the following sentence to the text: "Uncertainties in the differential slant columns due to uncertainties in the spectral calibration are  $\sim 5 \times 10^{13}$  molecules cm-2 for NO2 and  $\sim 2 \times 10^{15}$  molecules cm-2 for CH2O."

**P18, L10: Replace "more diffuse" with "nosier".**

Here we mean to describe the spatial features of CH2O as being more spread out (diffuse) than those of NO2, but realize this is not clear from the figures. We have removed this part of the sentence and now start the sentence with "Model profile shapes during the spirals…", and have added the reference to Fried et al. (2016b) which compared the P-3B with CMAQ.

**P19, L9: Please give a number for "a few percent".**

To address this comment and a related one by the other reviewer, we have added text and modified this sentence to read:

"When the spiral profiles are applied to the GCAS observations within their vicinity, the use of the observed profiles lowers the overall slope of GCAS tropospheric NO2 columns by 4 % (P-3B) and 2 % (Pandora) and the CH2O columns by 2 % (P-3B) as compared with coincident observations. The NO2 correlations with the P-3B and Pandora remain the same

but the correlation increases to r2 = 0.62 for P-3B CH2O. Individual coincident observations can change by as much as -50 to +35 % for NO2 (mean change of +1 +/- 10 %) and -15 to +25 % for CH2O (mean change of +3 +/- 8 %). The largest mean changes for a single day occur at the Deer Park site in the Pandora comparisons, where the GCAS NO2 column on 25 September is reduced by 15 % on average on average."

**P20, Figure 7: The figure is very crowded and error bars are not explained for GCAS and missing for P-3B (extrapolation errors?).**

These are precision uncertainties, which we forgot to mention. The extrapolation errors are difficult to quantify here (however, for NO2 they are the subject of another study by colleagues using the in situ ground-based network). We have added the following to the figure caption: "Error bars represent the uncertainty in the GCAS mean column due to retrieval noise from the observations used to calculate a mean column (typically several hundred at 250mx500m resolution); in the case of NO2, this uncertainty is generally negligible due to low relative error. Column precisions for P-3B observations are approximately 2 x 1013 molecules cm-2 (NO2) and 6 x 1013 molecules cm-2 (CH2O). Uncertainties from spatial variability and measurement accuracy are discussed in the text."

**P21, L 8-10: The sentence is very vague. Could you provide numbers?**

We have changed this sentence to: "More surprisingly, GCAS NO2 is 50 % (DS) and 33 % (MAX-DOAS) larger at high NO2 values." and also added "or from an underestimation of Pandora columns" to the next sentence as this was missing in the original text.

**3 Technical corrections**

P10, L5: maybe replace "absorber" with "trace gas"

P10, L7: add a "," after "atmosphere"

These have been changed in the text.

**Response to Referee #2**

We thank Referee #2 for helpful comments. We have made a number of changes to the paper based on these comments. In the following text, the referee's comments are in bold italics, followed by our response. The manuscript with changes tracked follows this response.

The paper reports results of the first intensive field campaign (Discover AQ, Texas 2013) of GCAS, a compact 2-channel airborne spectrometer. GCAS NO2 and HCHO retrievals are compared with trace gas columns derived from coincident in situ profile measurements made by instruments on a P-3B aircraft, and with NO2 observations from ground-based Pandora spectrometers, operating in direct sun and scattered light modes. In a previous paper, Nowlan et al. (2016), preliminary GCAS retrievals were compared with GEOTASO retrievals.

The paper provides a detailed overview of all retrieval steps and comparison with coincident measurements, including uncertainties, reasons for observed differences, etc. The paper contains useful content for the preparation of validation campaigns for the new generation of air quality sensors, such as TEMPO. These campaigns will likely involve GEOTASO, GCAS and profile flights for the airborne segment and Pandora spectrometers for the ground-based segment. The paper fits well within the scope of AMT, is well-written and well-structured. However, some revisions need to be conducted in the paper before publication.

**General comments:**

-What is a bit missing in the paper is a geophysical interpretation of the acquired data. Typically there were four similar flights per day over the area which provides a good view on the spatial distribution of NO2 and HCHO. Even if it is beyond the scope of this paper, a section discussing the changes in the trace gas field and possible explanations would be an added-value to the paper. There are a few sentences in Section 5 on this, but it could be more extended.

We have rearranged and expanded Section 5 to provide a better geophysical interpretation of the data. This section now includes a discussion of the meteorology during the campaign, which drove many of the differences in day-to-day and flight-to-flight observed trace gas fields. Please see the revised manuscript for the complete text.

-Secondly, it is not clear to me why profile shapes from CMAQ model output at 4 by 4 km are used for the AMF calculations, while you have NO2 and HCHO in-situ measurements from P-3B spiral flights available. I can understand this approach for the comparison with P-3B (Sect. 6.1) as you need independent data, but it is not clear why you don't use the P-3B profile shapes instead of CMAQ when comparing to PANDORA spectrometers (Sect. 6.2). I would expect that the CMAQ model does not represent the strong spatial variability of the 3D NO2 field, which you can expect in

**an urban area, something you mention as well in the manuscript (p.14, L.20). On the other hand, the dependency on the profile uncertainty was assessed in 6.1.3, and seems to be small.**

This paper describes the GCAS product on the NASA DISCOVER-AQ data archive, and this is the product we wish to validate in the comparisons with the P-3B and Pandora. We use the model to calculate all air mass factors in this product, although scattering weights are also included in case people want to use independent profiles (like the P-3B). We do this for consistency, as there are several stretches of the GCAS flight that do not coincide with P-3B profiles, days that do not have P-3B flights, Pandora sites with no P-3B profiles, and as the P-3B overpasses Houston 3x/day and GCAS typically 4x/day, there are many GCAS observations that do not satisfy our coincidence criteria with the P-3B of 1 hour. To address this and the other reviewer's comment about the P-3B comparisons themselves, we have expanded our section on using the P-3B for GCAS AMFs to include comparisons with Pandora, and made it into Section 6.3 for improved organization.

The text now reads: "In order to assess the dependence of the GCAS observations on the profile uncertainty, we also apply the P-3B profiles in place of model profiles in the GCAS AMF calculations and compare the new GCAS columns with the P-3B and Pandora columns. When the spiral profiles are applied to the GCAS observations within their vicinity, the use of the observed profiles lowers the overall slope of GCAS tropospheric NO2 columns by 4 % (P-3B) and 2 % (Pandora) and the CH2O columns by 2 % (P-3B) as compared with coincident observations. The NO2 correlations with the P-3B and Pandora remain the same but the correlation increases to r2 = 0.62 for P-3B CH2O. Individual coincident observations can change by as much as -50 to +35 % for NO2 (mean change of +1 +/- 10 %) and -15 to +25 % for CH2O (mean change of +3 +/- 8 %). The largest mean changes for a single day occur at the Deer Park site in the Pandora comparisons, where the GCAS NO2 column on 25 September is reduced by 15 % on average on average."

**Specific comments:**

P3, L10: Please add: Tack, F., Merlaud, A., Meier, A. C., Vlemmix, T., Ruhtz, T., Iordache, M.-D., Ge, X., van der Wal, L., Schuettemeyer, D., Ardelean, M., Calcan, A., Schönhardt, A., Meuleman, K., Richter, A., and Van Roozendael, M.: Intercomparison of four airborne imaging DOAS systems for tropospheric NO2 mapping – The AROMAPEX campaign, Atmos. Meas. Tech. Discuss., https://doi.org/10.5194/amt-2017-478, in review, 2018.

We have updated the references to include this paper.

**P4, L1: Please mention also explicitly the swath width at this altitude.**

We have added: "At a typical flight altitude of 9 km, this results in a swath width on the ground of about 6.7 km." When addressing this comment, we also changed the listed effective full FOV to 41 degrees, instead of the 45 degrees previously listed (which was the design FOV).

**P4, L12: Explain shortly the impact of not having a zenith sky reference measurement capability when compared to GEOTASO or ACAM.**

We have added the following sentence to the text: "As a result, the reference spectra required by the GCAS trace gas retrievals must be derived from nadir observations over clean areas with relatively low pollution."

**P4, L23: Maybe mention here already that GeoTASO and GCAS were intercompared in the previous paper.**

We have added the following sentence: "Preliminary GCAS and GeoTASO NO2 observations were compared in a previous paper (Nowlan et al., 2016)."

**P.5, L1: It would be interesting to mention the main drivers for the chosen flight path in section 3.1. Driven by Sources? Pandora sites?**

The flight paths were mainly driven by the location of existing surface air quality sites. We have added the following sentence to the beginning of Section 3: "Flight paths were chosen so that the aircraft passed over eight existing ground sites with surface air quality monitors several times per day, in support of the mission goal of investigating the relationship between trace gas columns and surface air quality."

**P.7, L.8: Please specify the vertical resolution (for the part below the aircraft).**

We have added the following sentence to this section: "The model's vertical resolution ranges from 22 m at the surface to ~200 m at an altitude of 2 km, further increasing to ~650 m by the aircraft flight altitude."

**P.8, L.10:* Note that the spectral performance can be affected by in-flight pressure changes as well: See Kuhlmann, G., Hueni, A., Damm, A., and Brunner, D.: An Algorithm for In-Flight Spectral Calibration of Imaging Spectrometers, Remote Sensing, 8, 1017, doi:10.3390/rs8121017, 2016.**

GCAS is backfilled with gaseous nitrogen and sealed prior to aircraft integration to mitigate moisture in the instrument. There may be small changes in internal pressure due to temperature differences during flight, but we expect the effects of wavelength changes due these pressure differences are likely fairly minimal compared with the thermal shifts in the instrument. To account for both pressure and thermal shifts, the wavelength shift for each observation is retrieved simultaneously with trace gas columns as described in Section 4. We have added the following to the section on spectral calibration: "Pressure changes within the instrument may also shift the wavelength calibration through changes in the index of refraction (Kuhlmann et al., 2016). These changes are minimized in GCAS and are primarily due to changes in ambient temperature as the instrument is backfilled with gaseous nitrogen and sealed prior to aircraft integration to mitigate moisture. The impact of wavelength shifts on retrievals is further minimized through simultaneous fitting of a wavelength shift for each observation as described in Section 4.2.3."

P.9. L13: Maybe I missed it but I could not find back the estimated reference spectrum amount. The estimation also affects the VCD uncertainty and should be mentioned in Section 4.7. You mention it as an uncertainty on P. 21, L6. We have added a discussion of the uncertainty from the reference in Section 4.7, and have added the following text to Section 4.3: "NO2 above the aircraft is dominated by stratospheric NO2 and varies primarily by time of day, and ranges between V 1 = 2.3-3.8 x 1015 molecules cm-2. CH2O in the model is more variable, with the early part of the campaign (4 to 18 September) seeing levels of V  $\uparrow$  = 2-25 x 1014 molecules cm-2 and the latter part seeing levels of V =  $1-3 \times 10^{14}$  molecules cm-2. For our chosen reference location, the modeled vertical columns below the aircraft are  $V_R \downarrow = 2.0 \times 10^{15}$  molecules cm-2 for NO2 and  $V_R \downarrow = 7.5 \times 10^{15}$  molecules cm-2 for CH2O. The modeled vertical columns above the aircraft at the reference location are  $V_R$   $\uparrow$  = 3.6 x 1015 molecules cm-2 for NO2 and  $V_R$   $\uparrow$  = 7.9 x 1014 molecules cm-2 for CH2O." and have also added the following to Section 4.3.2 for completeness: "The AMF is calculated scene-by-scene for each nadir observation. The reference spectrum AMFs at the swath center are  $A_R \downarrow = 1.65$  and  $A_R \downarrow = 1.92$  for NO2 and  $A_R \uparrow = 2.03$  and  $A_R \uparrow = 2.49$  for CH2O."

**P.9, L30: Why is the fitting window 420-465 nm for NO2 and for example not 425-490 nm as proposed by NDACC (for MAX-DOAS instruments)? Please clarify. On p. 3, L27 the wavelength range of the UV/Visible channel is 300-490 nm.**

The 420-465 nm fitting window is the same used by GeoTASO (Nowlan et al., AMT, 2016), which we have carried forward for GCAS for consistency. The lower limit is determined by the GeoTASO instrument, whose visible channel starts at 415 nm. The SNR on the first few wavelengths near the edge of the GeoTASO detector is typically low, so we chose 420 nm as a starting point for the retrieval. The 465 nm end point is that used by recent OMI retrievals (van Geffen et al., AMT, 2015). While it is possible to extent the fitting window to 490 nm with GCAS (or something slightly short of that to avoid using the edge of the detector where calibration and SNR can be somewhat poor), nadir viewing NO2 retrievals typically use a shorter wavelength cut-off to avoid complications from the surface, other absorbers or calibration effects. Richter et al. (AMT, 2011) did successfully extend the GOME-2 fit to 497 nm by including liquid H2O and sand in the retrieval. We attempted to use their fitting window of 425-497 nm with similar inputs, but the resulting GeoTASO NO2 retrievals were affected by large background biases (likely due to imperfect instrument calibration in combination with surface effects). As a result, we settled on 420-465 nm, which minimized unphysical biases while providing enough wavelengths for acceptable retrieval precision.

**P.11, L8: Please mention the spatial resolution of the MODIS BRDF product.**

We have added the following to the text: "This BRDF product is provided at a spatial resolution of 30 arcsec (~0.80 km in longitude by 0.92 km in latitude over Houston) every 8 days, based on 16 days of MODIS measurements."

**P.13, L.5: Please specify the impact of the spatial binning on the uncertainty for HCHO.**

We have added the text: "As a result, later in this paper we present CH2O maps at 1 km2 resolution, with an effective precision on the order of 7 x  $10^{15}$  molecules cm-2.". We also changed the caption in Figure 6 to be more specific and consistent with this number.

P. 13, L.32: The aerosol effect is also well explained in Meier, A. C., Schönhardt, A., Bösch, T., Richter, A., Seyler, A., Ruhtz, T., Constantin, D.-E., Shaiganfar, R., Wagner, T., Merlaud, A., Van Roozendael, M., Belegante, L., Nicolae, D., Georgescu, L., and Burrows, J. P.: High-resolution airborne imaging DOAS measurements of NO2 above Bucharest during AROMAT, Atmos. Meas. Tech., 10, 1831-1857, https://doi.org/10.5194/amt-10-1831-2017, 2017.

We have added this reference to others previously listed.

**P.14, L.12: Were there no aerosol instruments on the P-3B aircraft? In case not, it would be a real added-value for future campaigns, as you are already doing the efforts to perform spiral flights for NO2 and HCHO profiles. A good knowledge of the AOD profiles can indeed drastically decrease the uncertainties on the AMF.**

A suite of aerosol instruments was deployed on the P-3B aircraft, measuring aerosol dry size distribution, scattering/absorption coefficient and hydroscopicity. Probably even more useful for GCAS was the inclusion of the HSRL-2 lidar on the B-200 aircraft, along with GCAS. This was mentioned in Section 3.1 but we have now added to the sentence for clarity: "as well as the NASA High Spectral Resolution Lidar-2 (HSRL-2) instrument (Hair et al., 2008) for measuring aerosol profiles below the aircraft."

While much of the analysis of the Texas campaign is ongoing, the P3B and HSRL measurements have been used together in the following publication: Sawamura, P., Moore, R. H., Burton, S. P., Chemyakin, E., Müller, D., Kolgotin, A., Ferrare, R. A., Hostetler, C. A., Ziemba, L. D., Beyersdorf, A. J., and Anderson, B. E.: HSRL-2 aerosol optical measurements and microphysical retrievals vs. airborne in situ measurements during DISCOVER-AQ 2013: an intercomparison study, Atmos. Chem. Phys., 17, 7229-7243, https://doi.org/10.5194/acp-17-7229-2017, 2017.

We have performed a brief analysis using the HSRL AOD profiles to assess the uncertainties in the AMF from aerosols (discussed in Section 4.7.2) using some assumptions about optical properties. We are working on implementing more complete lidar measurements, including optical properties, in our radiative transfer code, and on a study to assess the effects of aerosols on the AMF in more detail using data from this and other campaigns.

**P.15,L.27:* Specify "near-surface" in order to avoid confusion with surface concentrations.**

We have changed the sentence to: "These figures illustrate the typical coverage of P-3B spirals relative to GCAS swaths, as well as the air mass measured in the bottom 2 km of the atmosphere by Pandora DS ground-based instruments."

**P.16, Figure3: Indication of the average wind direction (and speed) could help interpretation or describe in text if too variable (same for the other maps).**

We have added two paragraphs in Section 5 describing the wind patterns in the Houston area on the different days of the campaign, and further references that go into details of the meteorology.

**P.18, Figure 5: Please explain the reason for missing data. Sometimes a full across-track scanline is missing which is not related to cloud filtering. Are there other filters applied?**

These stripes occur when the flight software writes to disk. We have added the following line to the figure caption: "Periodic cross-track gaps in the data are due to write-to-disk intervals of the instrument. During these periods, the instrument does not acquire data thus producing small gaps in coverage."

**P.20, Figure 7b: Error bars not explained in text?**

We have added the following to the figure caption: "Error bars represent the uncertainty in the GCAS mean column due to retrieval noise from the observations used to calculate a mean column (typically several hundred at 250 m x 500 m resolution); in the case of NO2, this uncertainty is generally negligible due to low relative error. Column precisions for P-3B observations are approximately  $2 \times 10^{13}$  molecules cm-2 (NO2) and  $6 \times 10^{13}$  molecules cm-2 (CH2O). Uncertainties from spatial variability and measurement accuracy are discussed in the text."

**P.21, L.6: What about the use of the simple geometric AMF for PANDORA (P.6, L3). Could this also contribute (significantly) to the differences observed?**

The uncertainty from the use of the Pandora geometric AMF should be small (<1%, see Herman et al., JGR, 2009) at solar zenith angles less than 80°, so we don't expect its influence to be significant. There may be some influence from aerosols and clouds (see a similar discussion in https://www.atmos-meas-tech-discuss.net/amt-2018-57/amt-2018-57-AC3.pdf), but as both the Pandora data and GCAS data are cloud screened, this is likely minimal as compared to uncertainties introduced by other sources in the Pandora spectral fitting.

**Technical corrections:**

*P3, L1-L7: This paragraph is a bit too technical and would be better fitting in section 2.*

We have moved this paragraph to Section 2.

**P.15, L13: Change "individual industrial sites" to "single industrial sites" or "single Stacks".**

This has been changed to "single industrial sites and stacks".

**P.20, L13: Change line-of-site to line-of-sight**

This has been changed in text.

P.21, L27: Please reformulate as sentence is not clear. Maybe: "....with retrievals from a spectrometer from the Network for the Detection of Atmospheric Composition Change (NDACC) using..." + Please specify also the type of (DOAS) instrument. We have rearranged and shortened and these sentences to: "Previous comparisons of Pandora DS total column NO2 observations with other ground-based observations have shown good agreement (Herman et al., 2009;Wang et al., 2010a). In contrast, Knepp et al. (2017) compared a year of Pandora zenith-sky stratospheric NO2 slant columns with those from a zenith-looking UV-Vis spectometer (DOAS M07) from the Network for the Detection of Atmospheric Composition Change (NDACC) using different retrieval settings, and found Pandora underestimated the NDACC instrument by 7–40 %."

**P.22, L1: Please put comma after "inputs" and after "columns".**

These commas have been added.

**P.24, Figure10: acronym DS is used here for the first time. Should also be used in text**

**and written in full at first appearance.**

The acronym DS now replaces most uses of "direct sun" in text, and has been defined at first use.

The following is a tracked changes version of the manuscript. Note that latexdiff is unable to properly handle some section re-numbering -- these should appear correct in the manuscript without tracked changes.

**Nitrogen dioxide and formaldehyde measurements from the GEOstationary Coastal and Air Pollution Events (GEO-CAPE) Airborne Simulator over Houston, Texas**

Caroline R. Nowlan1, Xiong Liu1, Scott J. Janz2, Matthew G. Kowalewski2,3, Kelly Chance1, Melanie B. Follette-Cook2,4, Alan Fried5, Gonzalo González Abad1, Jay R. Herman6, Laura M. Judd7, Hyeong-Ahn Kwon8, Christopher P. Loughner9, Kenneth E. Pickering2,10, Dirk Richter5, Elena Spinei11, James Walega5, Petter Weibring5, and Andrew J. Weinheimer12 1Harvard-Smithsonian Center for Astrophysics, Cambridge, MA 02138, USA 2NASA Goddard Space Flight Center, Greenbelt, MD 20771, USA 3Goddard Earth Sciences Technology and Research, Universities Space Research Association, Greenbelt, Maryland, USA 4Morgan State University/GESTAR, Baltimore, MD 21251, USA 5Institute for Arctic and Alpine Research, University of Colorado, Boulder, CO 80303, USA 6University of Maryland, Baltimore County, Baltimore, MD 21201, USA 7NASA Langlev Research Center, Hampton, VA 23666, USA 8Seoul National University, Seoul, Republic of Korea 9NOAA 
[revised manuscript text omitted]
 3. NO2 is fit at wavelengths 420–465 nm with an NO2 absorption cross section at 294 K. The NO2 retrieval also simultaneously fits O3 at two temperatures, as well as H2O vapor and O2–O2, which all have spectral absorption features in the NO2 wavelength fitting window. The

30  $CH_2O$  retrieval is performed at 328.5–356.5 nm, and simultaneously fits NO2, O3, BrO and O2–O2. Both retrievals also fit the undersampling correction, Ring spectrum, a 5th-order scaling polynomial, and a 4th-order baseline polynomial. Each

retrieval also determines a wavelength shift that represents the relative difference in the detector pixel to wavelength registration between the radiance and reference spectra.

**4.3 Conversion to vertical column**

For air quality applications, we are interested in the vertical column density, V, of the absorber trace gas (NO2 or  $CH_2O$ ) in the troposphere. The vertical column density can be derived from the slant column density. S, using an air mass factor, A,

5 which describes the mean light path through the atmosphere, by

$$V = \frac{S}{A}.$$
(2)

In practice, the retrieval algorithm determines a differential slant column  $\Delta S$ , which is the difference between the slant column S of the absorber in the spectrum of interest, and the slant column  $S_R$  in the reference spectrum. Each of these slant columns is the sum of the slant column of absorber in the light path above  $(\uparrow)$  and below  $(\downarrow)$  the aircraft, so that

$$\Delta S = (S^{\downarrow} + S^{\uparrow}) - (S^{\downarrow}_{\mathrm{R}} + S^{\uparrow}_{\mathrm{R}}).$$
(3)

In terms of the air mass factor and vertical column, the vertical column below the aircraft can then be expressed as

$$V^{\downarrow} = \frac{\Delta S - V^{\uparrow} A^{\uparrow} + V_{\mathrm{R}}^{\downarrow} A_{\mathrm{R}}^{\downarrow} + V_{\mathrm{R}}^{\uparrow} A_{\mathrm{R}}^{\uparrow}}{A^{\downarrow}},\tag{4}$$

where the vertical columns  $V^{\uparrow}$ ,  $V_{\rm R}^{\downarrow}$  and  $V_{\rm R}^{\uparrow}$  are typically determined from a model. Because the flight altitude of 9 km is well above the majority of tropospheric NO2 and CH2O, we refer to  $V^{\downarrow}$  and  $V^{\uparrow}$  as the trospospheric and stratospheric trace 15 gas columns. NO2 above the aircraft is dominated by stratospheric NO2 and varies primarily by time of day, and ranges between  $V^{\uparrow} = 2.3 - 3.8 \times 10^{15}$  molecules cm-2. CH2O in the model is more variable, with the early part of the campaign (4) to 18 September) seeing levels of  $V^{\uparrow} = 2 - 25 \times 10^{14}$  molecules cm-2 and the latter part seeing levels of  $V^{\uparrow} = 1 - 3 \times 10^{14}$ molecules cm-2. For our chosen reference location, the modeled vertical columns below the aircraft are  $V_{\rm R}^{\downarrow} = 2.0 \times 10^{15}$ molecules cm-2 for NO2 and  $V_{\rm B}^{\downarrow} = 7.5 \times 10^{15}$  molecules cm-2 for CH2O. The modeled vertical columns above the aircraft 20 at the reference location are  $V_{\rm B}^{\uparrow} = 3.6 \times 10^{15}$  molecules cm-2 for NO2 and  $V_{\rm B}^{\uparrow} = 7.9 \times 10^{14}$  molecules cm-2 for CH2O.

**4.3.1 Air mass factor calculation**

We calculate the air mass factors on a scene-by-scene basis using the formulation of Palmer et al. (2001) and Martin et al. (2002) with the VLIDORT radiative transfer model (Spurr, 2006, 2008). In this approach, the radiative transfer model provides

25

10

scattering weights w as a function of altitude z. The scattering weights describe the sensivity of the measurement to the different altitude layers and are a function of the viewing geometry, ozone profile, aerosol and molecular scattering, and surface reflectance. These can be used with shape factor s, which is the normalized partial column n of the trace gas at at each altitude layer:

$$s(z) = \frac{n(z)}{\int_{z} n(z) \mathrm{d}z}.$$
(5)

The AMF is defined as

$$A = \int_{z} w(z)s(z)\mathrm{d}z.$$
(6)

The air mass factor below the aircraft  $A^{\downarrow}$  is calculated from the surface  $z_0$  to the aircraft altitude  $z_{ac}$  as

$$A^{\downarrow} = \int_{z_0}^{z_{ac}} w(z)s(z)\mathrm{d}z,\tag{7}$$

5 while the air mass factor above the aircraft  $A^{\uparrow}$  is determined from the aircraft altitude to the top of the atmosphere at  $z_{TOA}$ , with

$$A^{\uparrow} = \int_{z_{ac}}^{z_{TOA}} w(z)s(z)\mathrm{d}z.$$
(8)

**4.3.2 Radiative transfer calculations**

We use the radiative transfer algorithm to determine scattering weights in 56 vertical layers. These include the 45 CMAQ
layers up to ~19 km and 11 additional layers to 0.3 hPa. We use the MODIS BRDF (bidirectional reflectance distribution functions) gap-filled MCD43GF V005 Band 3 product (Schaaf et al., 2002; Sun et al., 2017) to represent surface reflectance in the VLIDORT model. This BRDF product is provided at a spatial resolution of 30 arcsec (~0.80 km in longitude by 0.92 km in latitude over Houston) every 8 days, based on 16 days of MODIS measurements. The MODIS Band 3 product is derived at 470 nm. While this is close to the NO2 fitting window, there currently exists no BRDF climatology at shorter wavelengths. We

15 determine effective BRDFs at 442 nm (NO2) and 342 nm (CH2O) by scaling the BRDF functions by the ratio of the  $0.5^{\circ} \times 0.5^{\circ}$  monthly OMI Earth Surface Reflectance Climatology product (OMLER) (Kleipool et al., 2008) at either 442 nm or 342 nm to its value at 470 nm. These results are typically within a few percent 2–3 % of the results derived using a black-sky/white-sky approach to estimate surface reflectance (McLinden et al., 2014).

Figure 2 shows profiles for 1) a sample polluted observation at the Moody Tower site in downtown Houston and 2) the

- 20 reference spectrum. For the AMF calculation, the shape factors are derived from the model profiles shown in Figures 2a and 2c and then applied to the corresponding scattering weights. Differences in the scattering weights of the reference and Moody Tower observations at higher altitudes are mainly driven by differences in the solar zenith angles. The smaller  $CH_2O$  scattering weights near the surface relative to those of  $NO_2$  indicate the relatively lower sensitivity of the observations to near-surface  $CH_2O$ . This is due primarily to the wavelength dependency of the AMF, as stronger Rayleigh scattering and ozone absorption
- at shorter wavelengths decreases the measurement sensivity to lower altitudes. The AMF is calculated scene-by-scene for each nadir observation. The reference spectrum AMFs at the swath center are  $A_{\rm R}^{\downarrow} = 1.65$  and  $A_{\rm R}^{\uparrow} = 1.92$  for NO2 and  $A_{\rm R}^{\downarrow} = 2.03$  and  $A_{\rm R}^{\uparrow} = 2.49$  for CH2O.